# Analyze Feature Flow to Enhance Interpretation and Steering in Language Models

**Daniil Laptev** [1 2]   **Nikita Balagansky** [1 2]   **Yaroslav Aksenov** [1]   **Daniil Gavrilov** [1]

## Abstract

We introduce a new approach to systematically map features discovered by sparse autoencoder across consecutive layers of large language models, extending earlier work that examined inter-layer feature links. By using a data-free cosine similarity technique, we trace how specific features persist, transform, or first appear at each stage. This method yields granular flow graphs of feature evolution, enabling fine-grained interpretability and mechanistic insights into model computations. Crucially, we demonstrate how these cross-layer feature maps facilitate direct steering of model behavior by amplifying or suppressing chosen features, achieving targeted thematic control in text generation. Together, our findings highlight the utility of a causal, cross-layer interpretability framework that not only clarifies how features develop through forward passes but also provides new means for transparent manipulation of large language models.

## 1. Introduction

Large language models (LLMs) excel at generating coherent text but remain largely opaque in how they store and transform semantic information. Previous research has revealed that neural networks often encode concepts as linear directions within hidden representations (Mikolov et al., 2013), and that sparse autoencoders (SAEs) can disentangle these directions into monosemantic features in the case of LLMs (Bricken et al., 2023; Cunningham et al., 2023). Yet, most methods analyze a single layer or focus solely on the residual stream, leaving the multi-layer nature of feature emergence and transformation underexplored (Balagansky et al., 2024; Balcells et al., 2024).

In this paper, we propose a data-free approach, based on

---

[1]T-Tech [2]Moscow Institute of Physics and Technology. Correspondence to: Nikita Balagansky <n.n.balaganskiy@tbank.ru>.

*Proceedings of the $42^{nd}$ International Conference on Machine Learning*, Vancouver, Canada. PMLR 267, 2025. Copyright 2025 by the author(s).

cosine similarity, that aligns SAE features across multiple modules (MLP, attention, and residual) at each layer, capturing how features originate, propagate, or vanish throughout the model in a form of "flow graphs".

1. **Cross-Layer Feature Evolution.** Using the pretrained SAEs that can isolate interpretable monosemantic directions, we utilize information obtained from cosine similarity between their decoder weights to track how these directions evolve or appear across layers. This reveals distinct patterns of feature birth and refinement not seen in single-layer analyses.

2. **Mechanistic Properties of Flow Graph.** By building a flow graph, we uncover an evolutionary pathway, which is also an internal circuit-like computational pathway, where MLP and attention modules introduce new features to already existing ones or change them.

3. **Multi-Layer Model Steering.** We show that flow graphs can improve the quality of model steering by targeting multiple SAE features at once, and also offer a better understanding of the steering outcome. This framework provides the first demonstration of such multi-layer steering via SAE features.

Our method helps to discover the lifespan of SAE features, understand their evolution across layers, and shed light on how they might form computational circuits, thereby enabling more precise control over model behavior.

## 2. Preliminaries

### 2.1. Linear representation hypothesis

To understand how models encode and process the information they learn, one can examine the geometric structure of their hidden representations and weights. Research has shown (Mikolov et al., 2013; Marks & Tegmark, 2024; Gurnee & Tegmark, 2024; Engels et al., 2025) that linear directions carry semantically meaningful information and may be used by models to represent learned concepts. Observations of this kind led to the development of the linear representation hypothesis, which can be stated as follows.

Hidden states $\mathbf{h} \in \mathbb{R}^d$ can be represented as sparse linear combinations of features $\mathbf{f} \in \mathbb{R}^d$ that lie in linear subspaces $\mathbb{F} \subset \mathbb{R}^d$. The impact of each feature is encoded by its magnitude $\|\mathbf{f}\|$. The total number of these linear subspaces with unique semantics greatly exceeds $d$, forcing the model to build an overcomplete basis in the feature space embedded in $\mathbb{R}^d$. During a forward pass, the model typically uses only a small fraction of them. These subspaces are usually one-dimensional lines, but more complex structures can appear (Engels et al., 2025).[1]

## 2.2. SAE and Transcoders

To retrieve such linear directions, Sparse Autoencoders (SAEs) (Bricken et al., 2023; Cunningham et al., 2023) were introduced. They decompose the model's hidden state into a sparse weighted sum of interpretable features.

Let $\mathcal{F}^{(P)} = \{\mathcal{F}_i^{(P)} \mid i \in \{1, ..., D\}\}$, where $D \gg d$ is the dictionary size, be a collection of one-dimensional features learned by an SAE at position $P$ in the model (e.g., after the MLP block). Then the SAE can be represented as

$$\mathbf{z} = \sigma(\mathbf{W}_{\text{enc}}\mathbf{h} + \mathbf{b}_{\text{enc}}),$$
$$\hat{\mathbf{h}} = \mathbf{W}_{\text{dec}}\mathbf{z} + \mathbf{b}_{\text{dec}},$$

where $\mathbf{W}_{\text{enc}}, \mathbf{b}_{\text{enc}}, \mathbf{W}_{\text{dec}}, \mathbf{b}_{\text{dec}}$ are SAE parameters, $\sigma(\cdot)$ is a nonlinear activation function, $\mathbf{h} \in \mathbb{R}^d$ is a model's hidden state, $\mathbf{z} \in \mathbb{R}^{|\mathcal{F}|}$ is the feature activation, and $\hat{\mathbf{h}}$ is the SAE's reconstruction of the hidden state.

Sparse autoencoders are usually trained to reconstruct model hidden states while enforcing sparse feature activations:

$$L = L_{\text{rec}}(\mathbf{h}, \hat{\mathbf{h}}) + L_{\text{reg}}(\mathbf{z}).$$

Typically, $L_{\text{rec}} = \|\mathbf{h} - \hat{\mathbf{h}}\|_2^2$, while $L_{\text{reg}}(\mathbf{z})$ is an $l_0$ proxy.

The choice of activation function $\sigma(\cdot)$ is crucial for achieving the desired representation properties. JumpReLU (Rajamanoharan et al., 2024) introduces a threshold parameter $\theta \in \mathbb{R}^{|\mathcal{F}|}$ that controls how large each pre-activation must be for the feature to become active:

$$\sigma(\mathbf{z}) = \mathbf{z}\, H(\mathbf{z} - \theta),$$

where $H$ is the Heaviside function.

Top-K (Makhzani & Frey, 2014; Gao et al., 2025), allows one to control the desired sparsity level by fixing $k$:

$$\sigma(\mathbf{h}) = \text{top}_k(W\mathbf{h} + b).$$

Instead of taking the top-$k$ per sample, BatchTopK selects the top $k \times b$ activations over all samples in the batch (Bussmann et al., 2024).

[1]There is a distinction between the weak and strong LRH. The strong version posits that there are *only* linear representations, while the weak version says that representations are *mostly* linear and one-dimensional.

Transcoders (Jermyn et al., 2024) are very similar to SAEs, but they reconstruct a different target. Typically, they are trained as interpretable approximations of MLPs:

$$\hat{\mathbf{h}}_{\text{post}} = \text{TC}(\mathbf{h}_{\text{pre}}),$$
$$L_{\text{rec}} = \|\mathbf{h}_{\text{post}} - \hat{\mathbf{h}}_{\text{post}}\|^2,$$

where $\mathbf{h}_{\text{pre}}$ is the pre-MLP hidden state, $\mathbf{h}_{\text{post}}$ is the post-MLP hidden state, and $\hat{\mathbf{h}}_{\text{post}}$ is the transcoder's prediction.

## 2.3. Features On Different Layers

Interconnections among SAE features trained on different layers of the same model have been reported and studied (Balagansky et al., 2024; Balcells et al., 2024; Ghilardi et al., 2024). Features in earlier layers tend to be low-level, often indicating word characteristics (e.g., words starting with certain letters), while features in later layers are typically more high-level and guide model behavior.

Sparse autoencoders are typically trained at three points in each layer: the output of the attention mechanism, the output of the MLP, and the residual stream. The latter is the main conduit of information within a transformer; MLP and attention modules read from it, process the data, and write their outputs back into it. According to Balagansky et al. (2024), most features in the residual stream remain relatively unchanged across layers. To identify similar features between different layers, one can define a permutation matrix $\mathbf{P}^{(A \to B)}$ that maps feature indices from layer $A$ to layer $B$, both having the same number of features $|\mathcal{F}|$:

$$\mathbf{P}^{(A \to B)} = \underset{\mathbf{P} \in \mathcal{P}_{|\mathcal{F}|}}{\arg\min} \sum_{i=1}^{d} \left\| \mathbf{W}_{\text{dec}_{i,:}}^{(B)} - \mathbf{W}_{\text{dec}_{i,:}}^{(A)} \mathbf{P}^{(A \to B)} \right\|^2,$$

where $\mathbf{W}_{(\cdot)}^{(A)} \in \mathbb{R}^{d \times |\mathcal{F}|}$ is a parameter of the SAE trained on the residual stream after layer $A$, and $\mathcal{P}_{|\mathcal{F}|}$ is the set of permutation matrices of size $|\mathcal{F}| \times |\mathcal{F}|$.

Dunefsky et al. (2024) finds a computational graph through the MLP layers by training transcoders:

$$\mathbf{z}(\mathbf{h}_{\text{pre}})_i \big( \mathbf{W}_{\text{dec}}^{(A)\intercal} \mathbf{W}_{\text{enc}}^{(B)\intercal} \big)_{i,:}. \tag{1}$$

Here, $\mathbf{W}_{\text{dec}}^{(A)\intercal} \mathbf{W}_{\text{enc}}^{(B)\intercal} \in \mathbb{R}^{|\mathcal{F}| \times |\mathcal{F}|}$ serves as a transition operator between the feature spaces of layers $A$ and $B$, revealing which features in $B$ are ancestors for the $i$th feature in $A$.

Matrices $\mathbf{P}^{(A \to B)}$ and $\mathbf{W}_{\text{dec}}^{(A)\intercal} \mathbf{W}_{\text{enc}}^{(B)\intercal}$ are in some sense similar. We explore this further in Appendix F.

## 3. Method

### 3.1. Motivation

Although SAEs provide human-interpretable features, they do not explain how these features interact or how the

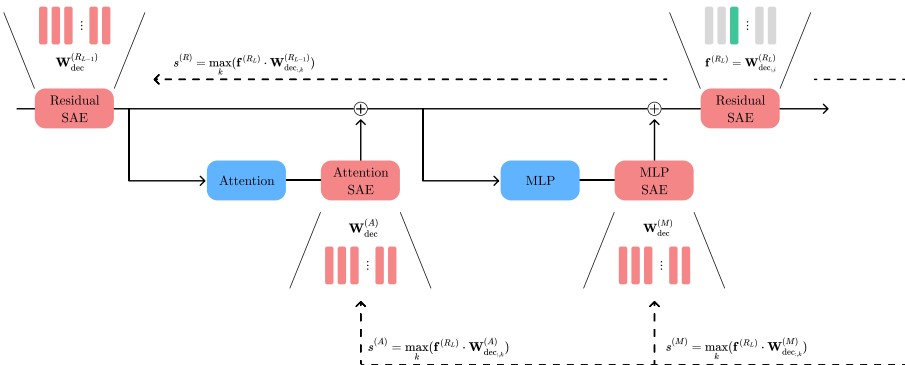

*Figure 1.* Schematic illustration of inner-layer matching. We select a feature with index $i$ on the SAE trained at the layer output. Its embedding $\mathbf{f}$, which is the $i$th column of this SAE's decoder weight, is compared to every column of other SAEs on the same layer (after the MLP and attention blocks, as well as with the SAE on the residual stream before some layer). These comparisons indicate the feature's source. See Section 3.3 for more details.

model's computation is carried out. Understanding this is crucial for more precise model manipulation.

A key principle is that such understanding can be obtained by linking features at different levels of a model (Balagansky et al., 2024; Dunefsky et al., 2024). If we want to find features shared by two SAEs trained at positions $A$ and $B$, we need to discover a mapping

$$\mathbf{T}^{A \to B} : \mathcal{F}^{(A)} \to \mathcal{F}^{(B)}.$$

This drives methods (Balagansky et al., 2024; Balcells et al., 2024) for finding these shared features and architectures (Lindsey et al., 2024) that ensure persistent collections of features by design.

By grouping similar features, we can find those that remain the same across different positions (by repeatedly applying mapping rules) or uncover those unique to specific points in the model. This helps us understand how semantic structure and computational modes evolve, while SAE features serve as an interpretable proxy.

### 3.2. Feature matching

Several methods exist for matching features between layers and modules. One approach uses correlations between activations (Wang et al., 2025; Balcells et al., 2024), but it requires considerable data to compute activation statistics. Another is a data-free approach based on SAE weights (Dunefsky et al., 2024; Balagansky et al., 2024). We found that cosine similarity between decoder weights is a valuable similarity metric, and we focus on this approach.

Let $\mathbf{f} \in \mathbb{R}^d$ be the embedding of some feature $\mathcal{F}_i^{(A)}$, trained

at position $A$. This vector is the $i$th column of $\mathbf{W}_{\text{dec}}^{(A)}$. Also let $\mathbf{W}_{\text{dec}}^{(B)} \in \mathbb{R}^{d \times |\mathcal{F}|}$ be the decoder weights of an SAE trained at position $B$. We find the matched feature index as

$$j = \arg\max_k \big(\mathbf{f} \cdot \mathbf{W}_{\text{dec}_{:,k}}^{(B)}\big).$$

Then we say that $\mathcal{F}_i^{(A)}$ corresponds to $\mathcal{F}_j^{(B)}$. We assume that both $\mathbf{f}$ and the columns of $\mathbf{W}_{\text{dec}}^{(B)}$ have unit norm.

More generally, we define

$$\mathbf{T}^{(A \to B)} = \mathbb{I}_{x>0}\big(\text{top}_k\big(\mathbf{W}_{\text{dec}}^{(A)\mathsf{T}} \mathbf{W}_{\text{dec}}^{(B)}\big)\big),$$

where $\mathbb{I}_{x>0}$ is an indicator function and $\text{top}_k(\cdot)$ zeroes out values below the $k$th order statistic. When $k = 1$, this many-to-one matching extends the one-to-one approach in Balagansky et al. (2024). Although top-$k$ handles many-to-many cases, we focus on many-to-one as a substantial extension of previous work.

This technique assumes SAEs are trained on hidden states whose structure is aligned. For instance, Gemma Scope (Lieberum et al., 2024) attention SAEs are trained before a nonlinear transformation at dimension 2048, whereas MLP and residual SAEs are trained on dimension 2304, so our method cannot be applied there. As shown in Section 5.1, the data distribution can also affect these results.

### 3.3. Tracking the evolution of feature

There are four main computational points in a standard transformer layer: the layer output $R_L$, the MLP output $M$, the attention output $A$, and the previous layer output $R_{L-1}$

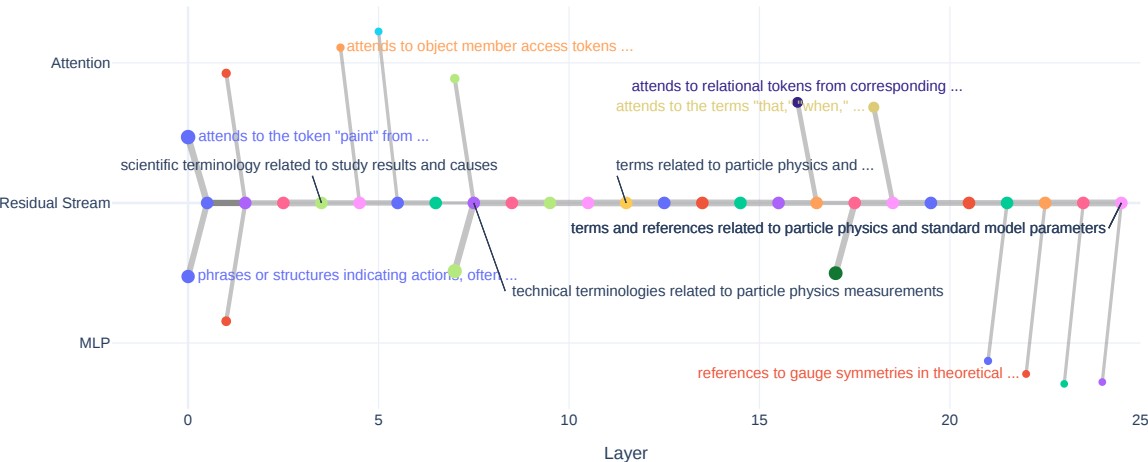

*Figure 2.* An illustration of the resulting flow graph, which we also use in the deactivation experiment (section 5.2). As a starting point, we select the feature on the 24th-layer residual with index 14548. For a detailed explanation of this graph, see Appendix E.

(the layer input). MLP and attention modules read from $R_{L-1}$ and their outputs produce $R_L$.

We pick a feature from the SAE trained on $R_L$ with embedding $\mathbf{f} \in \mathbb{R}^d$. Let $\mathbf{W}_{\text{dec}}^{(P)} \in \mathbb{R}^{d \times |\mathcal{F}|}$ for $P \in \{M, A, R \equiv R_{L-1}\}$ be the corresponding decoder weights. We compute the similarity between the target feature and $P$ as the maximum cosine similarity over the columns of $\mathbf{W}_{\text{dec}}^{(P)}$:

$$s^{(P)} = \max_k \big( \mathbf{f} \cdot \mathbf{W}_{\text{dec}_{:,k}}^{(P)} \big),$$

as illustrated in Figure 1. From these scores, we can infer how the feature relates to the previous layer or modules:

A) High $s^{(R)}$ and low $s^{(M)}, s^{(A)}$: The feature likely existed in $R_{L-1}$ and was *translated* to $R_L$.

B) High $s^{(R)}$ and high $s^{(M)}$ or $s^{(A)}$: The feature was likely *processed* by the MLP or attention.

C) Low $s^{(R)}$ but high $s^{(M)}$ or $s^{(A)}$: The feature may be *newborn*, created by the MLP or attention.

D) Low $s^{(R)}$ and low $s^{(M)}, s^{(A)}$: The feature cannot be easily explained by maximum cosine similarity alone.

Thresholds for "high" and "low" are specific for each layer.

We use a backward-matching approach because it naturally answers, "Where did this feature come from?" Forward-matching answers, "Where does this feature go?" but is less helpful for finding novel or transformed features.

**Long-range feature flows.** As we progress through the model, semantics undergo substantial changes, making direct long-range matching challenging. We address this by performing short-range matching in consecutive layers and composing the resulting transformations. For a given feature, we construct a flow graph from the initial layer to the final layer. This flow graph traces a path that reveals how the feature's semantic properties evolve. An example of such a graph is presented in Figure 2.

Currently, individual SAE features or their groups (Engels et al., 2025) are treated as units for study. However, we believe that these flow graphs may also become a compelling area for future research.

### 3.4. Identification of linear feature circuits

Model behavior can be decomposed into computational subnetworks, called *circuits*, which perform task-specific operations (Elhage et al., 2021; Marks et al., 2025). Our method helps identify potential circuits where MLP and attention modules add or remove features in a mostly linear way. High values of $s^{(M)}$ or $s^{(A)}$ are strong indicators of these circuits. We validate this in our experiments, focusing on how a feature's meaning evolves. Examples appear in Appendix E.

### 3.5. Model steering

Flow graphs can also help steer the model toward desired behaviors by identifying feature sets we want to manipulate. By carefully selecting them, one can preserve both alignment and core model capabilities, and our method facilitates

discovery of such feature groups. By examining flow graphs built from those features, one can better understand and predict the behavior of the model after steering. Section 5.3 and Appendix B illustrate this process.

## 4. Experimental Setup

### 4.1. Models and SAEs

We conduct our main experiments with the Gemma 2 2B model (Gemma Team, 2024) and the Gemma Scope SAE pack (Lieberum et al., 2024) with a JumpReLU activation function and dictionary size of 16k features. We also test our approach on LLama Scope (He et al., 2024) (see Appendix D), which was trained with TopK activation function and was converted to a JumpReLU after training. In addition, we train our own JumpReLU SAEs for the attention output (before it is added back to the residual stream) on every layer of the Gemma model, following the Gemma Scope training pipeline.

We obtain interpretations from Neuronpedia[2], which also serves as an additional evaluation tool. Interpretations for newly trained attention features were not available, and none were provided for LLama Scope.

### 4.2. Overview of experiments

We design our experiments to analyze how residual features emerge, propagate, and can be manipulated across model layers. Specifically, we aim to: (i) determine how features originate in different model components, (ii) assess whether deactivating a predecessor feature truly deactivates its descendant, and (iii) use these insights to steer the model's generation toward or away from specific topics.

Below is a concise summary of each experiment. See Appendices A and B for detailed setup.

**Identification of feature predecessors.** We first verify that cosine similarity relations used for single-layer analysis align with actual activation correlations. A target feature in the residual stream $R_L$ is matched with the previous residual $R_{L-1}$, the MLP output $M$, or the attention output $A$ features. If none are active, we label it "From nowhere." By applying this process on four diverse datasets, we confirm the above-stated relation, and we also analyze how these groups are distributed across layers.

**Feature Deactivation.** We measure causal relationships by intervening on hidden states: if deactivating a predecessor also deactivates target feature, we infer a causal link.

Given hidden states $\mathbf{h}$ at the predecessor's position (previous residual, MLP, or attention output), we apply transformation

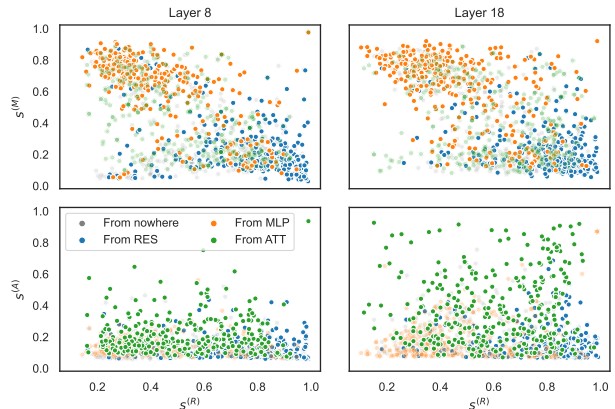

*Figure 3.* Example of cosine similarity vs. simultaneous activation with a predecessor (350 features were sampled per layer). "From MLP" and "From RES" groups are notably different: high $s^{(M)}$ and low $s^{(R)}$ suggest simultaneous activation with an MLP-module match. Cosine similarity serves as a good proxy for shared semantic and mechanistic properties.

$\mathbf{h} \leftarrow \mathbf{h} + a(r-1)\mathbf{v}$, where $a$ is the predecessor's activation strength, $\mathbf{v}$ its embedding, and $r$ a rescaling coefficient ($r = 0$ for deactivation). We expect this to remove the feature from the hidden state, preventing further propagation.

We evaluate four matching strategies: (1) *random* sampling from top-5 cosine-similarity matches, (2) *permutation-based* (Balagansky et al., 2024), (3) $\text{top}_1$ cosine similarity (our method), and (4) $\text{top}_5$ cosine similarity where all five matched features must be inactive to treat this predecessor as inactive. Effectiveness is quantified via *successful deactivation rate* and *activation change* (higher when new strength approaches 0).

**Model Steering.** We test whether multi-layer feature activation/deactivation can control theme generation. For a target topic, we intervene on relevant features across layers and assess text quality.

As a baseline we use initial features from which we build flow graphs. We compare *single-layer* (layer $l$ only) and *cumulative* (layers $0$ to $l$) interventions, applying the same rescaling for deactivation. For activation, we add scaled embeddings. Multi-layer strategies include *linear* and *exponential* decay of steering coefficients with respect to the layer index, and *constant* scale for all layers (Appendix B).

We measure *Behavioral* (topic presence) and *Coherence* (language quality) scores, and use their product as final metric (for deactivation $(1 - \text{Behavioral}) \times \text{Coherence}$).

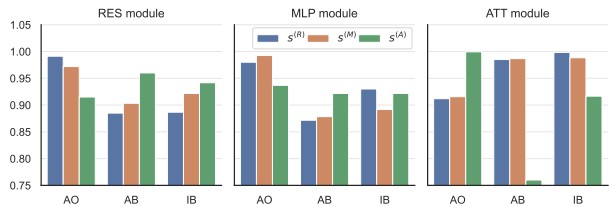

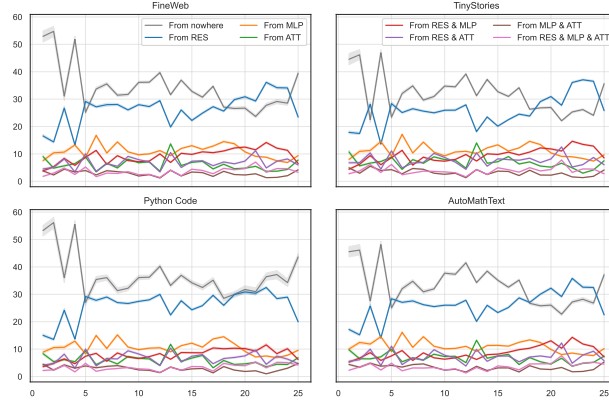

*Figure 4.* Percentage of statistically significant differences between groups for each module's similarity scores. AO means module $P$ is active in only one group, AB means active in both, and IB means inactive in both. For MLP, two groups differ in $s^{(R)}$ only 87% of the time when MLP is active in both groups.

*Figure 5.* Percentages of each group at each layer of Gemma 2 2B, illustrating how feature formation proceeds in the model.

## 5. Results

### 5.1. Identification of feature predecessors

In this experiment, we validate the single-layer analysis patterns from Section 3.3 by checking when target residual features and their predecessors activate simultaneously. For each activated residual feature, we assign it to a group based on which predecessors are also active. For example, if both the previous residual and MLP predecessors are active, the feature is categorized as "From RES & MLP." We then examine the distributions of scores within these groups.

Figure 3 reveals visually distinct score distributions across different groups. We quantify these differences with a Mann-Whitney U test on every pair of groups, for each dataset and layer, and then compute the fraction of tests with $p < 0.001$.

We observe that two groups may differ with respect to $s^{(P)}$ if module $P$ is active only in one group (and indistinguishable if $P$ is active or inactive in both groups). For example, "From MLP" and "From MLP & ATT" differ by $s^{(R)}, s^{(M)}, s^{(A)}$ in 67%, 72%, and 100% of tests, respectively. Figure 4 shows the total percentage of passed tests.

Figure 5 shows how these groups spread across layers, suggesting conceptual formation in earlier layers. From layers 0–5, "From nowhere" and "From RES" may reflect a high-entropy, early-stage process that stabilizes by about layer 5. After layer 18, where we see a bump for "From MLP", fewer new features emerge, and most features propagate from preceding layers.

There is also a three-part partition in the distribution of groups: approximately [0, 5] where uncertainty dominates, [6, 15] with somewhat stable dynamics, and [16, 25] where "From RES" group presence starts to rise and "From MLP" group diminishes after layer 18, implying that fewer new features appear in later layers.

We observe differences between datasets in the latter layers. The Python code dataset contains the least amount of natural

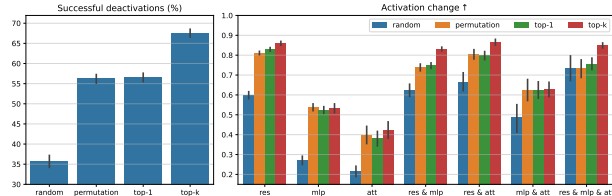

*Figure 6.* Deactivation methods compared. Group labels show which active predecessors were deactivated. The random approach underperforms, suggesting that choosing the $\text{top}_1$ feature is already meaningful for causal analysis.

language, and TinyStories has the most natural and simple language structure. The rarity of groups with activated attention could stem from our SAE training rather than an inherent property of Gemma. However, in the LLama Scope case (Figure 19(b)), we observe a slightly similar pattern, which indicates that this is indeed the property they share.

We have observed that group identification performance is on par with Pearson correlation-based matching methodology. The latter reduced the "From nowhere" group presence, but did not consistently outperform our method and performed worse on out-of-distribution Python code. See more details in Appendix C.4.

### 5.2. Deactivation of features

We compare the $\text{top}_1$ approach (choosing the most similar predecessor by cosine similarity) with randomly picking one of the $\text{top}_5$ candidates. Figure 6 shows that the random method sharply reduces deactivation success, confirming that $\text{top}_1$ is informative for causal analysis.

For MLP and attention predecessors, $\text{top}_1$ and $\text{top}_5$ per-

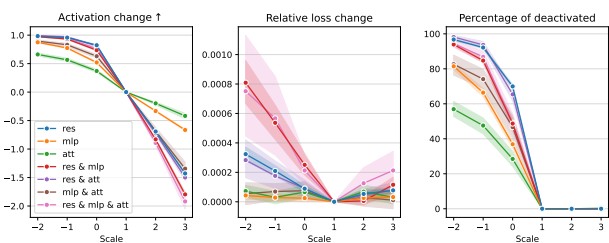

| | Mean AC | Success rate |
|---|---|---|
| Top-1 Cosine | 0.75 | 65% |
| Top-1 Pearson | 0.74 | 65% |
| Exhaustive Search | 0.83 | 73% |

*Table 1.* Comparison of deactivation methods. The exhaustive search evaluates all activated predecessor features individually and reports maximum performance. The similar results between correlation-based and our data-free method validate our approach.

*Figure 7.* Impact of different $r$ values on deactivation success, with rescaling of all available predecessors. When $r < 1$, the activation change grows nonlinearly, indicating alternative causal pathways still convey information. Relative loss change measured as $(L_{\text{new}} - L_{\text{old}})/L_{\text{old}}$ is a proxy for forward pass impact.

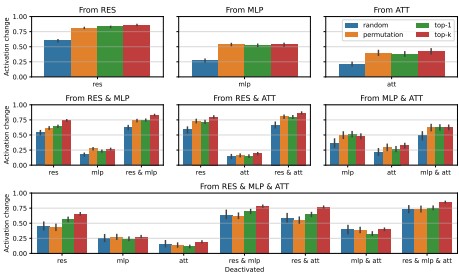

*Figure 8.* Mean activation changes when deactivating one predecessor at a time. Deactivation of some predecessor causes less impact if this predecessor is not activated alone, which leads to the conclusion that combined groups exhibit circuit-like behavior.

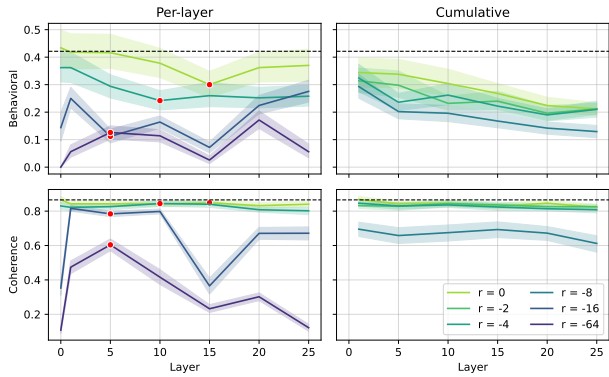

*Figure 9.* Deactivating the "Scientific concepts and entities" theme. The dashed black line shows the default generation score. Red points mark the best layer for each $r$ in the single-layer method. Larger $r$ boosts performance but shifts the optimal layer earlier.

form similarly. Differences arise mainly when a residual predecessor combines with another module, indicating that we might miss other types of causal relations.

Finally, we vary the rescaling coefficient $r$ to see how it affects deactivation results (Figure 7). Different groups react differently to rescaling. Positive rescaling (boosting active features) matters most when residual features mix with MLP or attention. Negative rescaling most strongly affects "From RES." Reducing "From RES & MLP" or "From RES & MLP & ATT" increases the loss change more than reducing "From RES" alone, highlighting MLP's critical role in these circuit-like interactions.

Figure 8 further shows that deactivating a single predecessor causes a greater activation strength drop if it is a group with a single predecessor, which may indicate circuit-like behavior in combined groups.

To compare our method with optimal performance, we test three approaches: (1) top-1 cosine similarity matching, (2) top-1 Pearson correlation matching, and (3) an exhaustive search for maximum achievable performance. The search procedure deactivates each active predecessor feature individually, with activation change computed only for target

features identified by either cosine or Pearson matching as "From RES", "From MLP", or "From ATT" to ensure fair comparison and computational feasibility. Testing 1,894 features across two layers (each deactivated via all three methods) yields the results in Table 1, showing comparable performance between cosine and Pearson methods, additionally validating our data-free approach.

### 5.3. Model steering

To evaluate interventions based on flow graphs, we use them to suppress or activate topics in generation. Figure 10 demonstrates that our method identifies more effective steering features across layers compared to single-feature interventions on the initial feature set. The cumulative approach additionally provides two key advantages: (1) reduced sensitivity to hyperparameter choices, and (2) improved performance with smaller hidden state perturbations.

Figure 9 analyzes the impact of rescaling coefficient $r$ on deactivation effectiveness. We observe that larger $r$ values shift the optimal intervention point toward earlier layers, while smaller $r$ values distribute the intervention effect more evenly across the network depth.

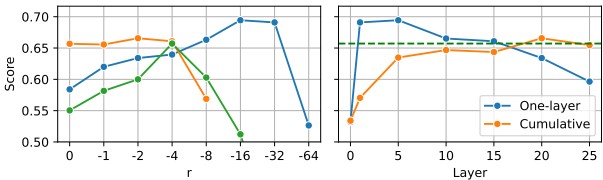

*Figure 10.* Comparison of best deactivation scores. The green line indicates deactivation using only the initial feature set. Interventions on layers detected by our method (orange, blue) perform better across different $r$ values, suggesting additional discovered features reduce hyperparameter sensitivity.

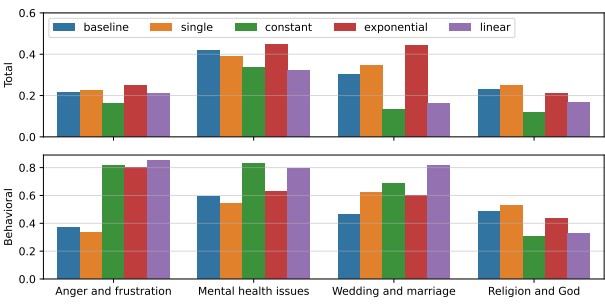

*Figure 11.* Activation of specific topics. We compare single-layer steering and cumulative approaches with three rescaling strategies (Appendix B). Activating multiple similar features amplifies a topic's presence but may degrade overall text coherence.

Figure 10 shows that cumulative intervention outperforms the single-layer approach in a low $r$ regime, suggesting that small interventions distributed over multiple layers may be more effective for controllable generation.

For activation tasks, we boost topic presence by activating multiple similar directions. Figure 11 shows that cumulative methods typically strengthen the topic signal but can reduce text quality. In some cases, the effect is clear: steering a feature tied to "Religion and God" can shift outputs toward biblical text, and if we examine the flow graph for that feature, we see that earlier layers are indeed linked to it.

## 6. Discussion

**Identification of feature predecessors.** Our results indicate that (i) similarity of linear directions is indeed a good proxy for activation correlation, and (ii) the structure of these groups differs across layers, possibly reflecting the properties of information processing within the model.

We suspect that "From MLP," "From ATT," and "From MLP & ATT" primarily contain *newborn* features introduced at their respective layers, whereas groups that combine the

residual stream with a module tend to hold *processed* features. The decline of "From MLP" and the rise of "From RES" groups shown in Figure 5 may indicate that later layers form fewer new features than intermediate layers.

**Deactivation of features.** We (i) confirm that $\text{top}_1$ similarity provides valuable information about causal dependencies, and (ii) conclude that groups respond differently to the deactivation of certain predecessors, indicating that they have distinct mechanistic properties and may exhibit circuit-like behavior. The fact that residual predecessors are the most influential could be explained by the nature of the residual stream as the main communication channel, so removing the feature at module will not prevent it from further propagation if it already exists in the residual stream.

**Model steering.** If we want to reduce a particular feature at inference, we typically adjust its magnitude. However, achieving a significant reduction may require large adjustment scales, which can alter the distribution of hidden states. Because we know which features contribute to the appearance of the feature we want to reduce, we can also adjust those. From this perspective, it is possible to make multiple smaller adjustments rather than one large one, avoiding dramatic changes to the overall distribution of hidden states.

Flow graphs may help to understand the effect of steering and identify related features, but the downstream result depends on the properties of the specific graph. Overall, we conclude that they allow to find impactful features for intervention. We hypothesize that removing topic-related information early allows later layers to recover general linguistic information, aligning with the ability of LLMs to "self-repair" after "damaging" the information processing flow by pruning or, in our case, intervention into the structure of hidden states (McGrath et al., 2023).

Overall, our method provides a straightforward way to identify and interpret the computational graph of the model without relying on additional data, achieving performance similar to Pearson correlation matching; the resulting graph can then be used for precise control over the model's internal structure. To the best of our knowledge, we are the first to use SAE features from different layers to control LLM generation. We believe that this work opens a new perspective for zero-shot steering.

## 7. Related Work

Multiple works have investigated feature circuits in language models. Conmy et al. (2023) proposed pruning connections between modules that do not affect the output. Ge et al. (2024) suggested using gradients to decide whether to prune connections between modules; they also demonstrated that their method can be used to find circuits on the feature

level with skip SAE, which is equivalent to transcoders. Dunefsky et al. (2024) showed that circuits can be found without a backward pass, relying solely on activations and transcoders' weights. Balagansky et al. (2024); Balcells et al. (2024) studied feature dynamics in the residual stream during the forward pass; however, these works focus exclusively on residual stream features and do not investigate the properties of the resulting computational graph or its application to steering. Additionally, SAE features as steering vectors were explored in Chalnev et al. (2024), but their approach is data-dependent and does not involve a multi-layer steering procedure. In contrast, our work advances these findings by introducing a straightforward and interpretable data-free method for multi-layer steering, which also enables the tracking of concept evolution across layers and the identification of computational circuits through targeting the weights of pretrained SAEs.

## 8. Conclusion

In our work, we propose using SAEs trained on different modules and layers of the base model to find a computational graph consisting of SAE features. Through our experiments, we validate that these graphs can describe most of the feature dynamics. Finally, we show that such graphs can be used for steering model behavior, thereby improving steering of LLMs with SAEs.

Advancements in model steering suggest focusing on more sophisticated steering approaches. For example, while we can reconstruct feature predecessors from multiple blocks in previous layers, it is evident that features are somewhat tangled across layers (when reducing the magnitude of a predecessor feature, all subsequent computations change). Thus, it may be helpful to concentrate on disentangling these connections across different layers. Other directions for better steering could also exist, thus opening new possibilities on further enhancing LLM controllable generation.

## Impact statement

Our work offers a method to systematically identify and manipulate latent features in large language models, thereby advancing the field of controllable generation. This improved controllability has positive implications for alignment, interpretability, and safe deployment of AI systems, as it can allow developers to steer models away from harmful or biased outputs. At the same time, similar techniques could be repurposed for unsafe or malicious behavior by those aiming to bypass safeguards or exploit hidden model pathways. These dual-use concerns highlight the importance of continued research and open discussion on controllable generation, rather than a cessation of study. By deepening our collective understanding, we are better equipped to de-

velop robust norms, policies, and technical safeguards that promote beneficial applications while mitigating the risks of misuse.

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

# A. Detailed Experimental Setup

## A.1. Identification of feature predecessors

This experiment aims to validate our proposed approach for determining the origin of a feature. Specifically, we verify whether the cosine similarity relations described for single-layer analysis align with the correlation between the features' activations. For a target feature from $R_L$, we consider it to originate from $R_{L-1}$ if the matched feature on $R_{L-1}$ is active while the matched features on $M$ and $A$ are inactive. There are seven possible combinations of activated predecessors; if none of these is active, the feature is assigned to an eighth group, "From nowhere."

We use four datasets for this analysis: FineWeb (Penedo et al., 2024) (general-purpose texts), TinyStories (Eldan & Li, 2023) (short synthetic stories), AutoMathText (Zhang et al., 2024) (math-related texts), and PythonGithubCode[3] (pure Python code). From each dataset, we select 250 random samples; for each sample, we pick 5 random tokens (excluding the BOS token). We then iterate over every activated feature on every layer and determine its group (i.e., which predecessor combination leads to that feature's activation).

## A.2. Deactivation of features

To further validate the proposed method, we measure the causal relationship between a feature and its predecessor by intervening directly in the model's hidden state. Specifically, we deactivate the predecessor by removing its corresponding decoder column from the relevant hidden state (i.e., at the MLP output, attention output, or previous layer output). We expect that deactivating the matched predecessor feature will also deactivate the target feature (at the end of the layer).

**Feature rescaling.** Consider a hidden state $\mathbf{h}_t \in \mathbb{R}^d$ for a specific token $t$. Suppose we want to modify the strength of $f$ features within this hidden state. Let $\mathbf{V} \in \mathbb{R}^{d \times f}$ be the embeddings of these $f$ features, and let $\mathbf{a}_t \in \mathbb{R}^f$ be their activation strengths for token $t$. We define rescaling as:

$$\mathbf{h}_t \leftarrow \mathbf{h}_t + (r-1)(\mathbf{a}_t \cdot \mathbf{V}^\mathsf{T}),$$

where $r$ is the rescaling coefficient. This method also allows us to rescale a feature to a desired strength for steering. We refer to rescaling as *positive* when $r \geq 1$, and *negative* otherwise.[4]

In the context of SAEs, we approximate hidden states with a linear combination of feature decoder columns (plus a bias term that does not depend on activation strength, and is therefore omitted). Setting $r = 0$ removes the selected features from the existing linear combination, which is (up to SAE reconstruction error) the same as setting those features' activations to zero.

**Experimental protocol.** In this experiment, we apply the above transformation only to the specific token where we detect the residual feature. We select 35 texts from FineWeb, choose 5 random tokens per text, and focus on layers 6, 12, and 18. For each layer–token pair, we randomly sample up to 25 features and deactivate them if they do not belong to the "From nowhere" group.

To assess the effectiveness of deactivation, we compare four matching methods:

- *permutation*: Deactivate the predecessor feature identified by permutation,

- $\text{top}_1$: Deactivate the most similar predecessor feature (based on cosine similarity of decoder embeddings),

- $\text{top}_k$ ($k = 5$): Deactivate the five most similar predecessor features,

- *random*: Randomly choose one from the top five most similar features.

The $\text{top}_1$ method is our main focus. For each method, we first identify the group of the target feature and then perform the deactivation. For the $\text{top}_5$ method, we consider the predecessor active if at least one of the 5 selected features is active.

We evaluate two main metrics:

---

[3] https://huggingface.co/datasets/tomekkorbak/python-github-code
[4] This is essentially equivalent to the method discussed in Templeton et al. (2024), see section "Methodological details."

| Feature index | Interpretation from Neuronpedia |
|---|---|
| 3/res/9811 | terms related to gravity and its influences |
| 18/res/14053 | terms related to theoretical frameworks and conceptual models |
| 18/res/1336 | references to Dark Matter and astronomical phenomena |
| 20/res/4506 | terms related to physical laws and scientific principles |
| 21/res/13226 | references to quantum concepts and theories |
| 22/res/9002 | terms related to models and their specifications, |
| 22/res/15105 | terms related to force and energy dynamics |
| 23/res/4086 | terms related to forces and dynamics in physical systems |
| 24/res/7017 | terms related to electromagnetic interactions and properties |
| 24/res/14548 | terms and references related to particle physics and standard model parameters |

*Table 2.* Features initially chosen for deactivation of "Scientific concepts and entities" theme.

- **Successful deactivations**: The number of times a feature was deactivated, divided by the number of times it had an active predecessor.

- **Activation change**: Defined as $1 - \left(\mathbf{z}_i^{\text{new}}/\mathbf{z}_i^{\text{old}}\right)$ for target feature $i$. This metric equals 1 when the feature is fully deactivated, and can be interpreted as a measure of causal dependency between predecessor and target features.

## B. Steering details

To further test whether our feature-matching approach enables effective model steering, we design a procedure to either suppress or promote particular themes in the generated text. We begin by identifying a small set of features for each theme, guided by Neuronpedia entries. We then build flow graphs (from layer 0 to layer 25) to trace how theme-related features evolve across the network. If the semantic meaning of a feature remains consistent and relevant, we add it to our target collection; otherwise, we continue searching until we have a satisfactory set of features.

We compare a *single-layer* steering strategy (affecting only the features of one layer) to a *cumulative* strategy (affecting features from the current and all previous layers) to see whether distributing interventions across the network can produce more robust thematic shifts. We also measure any corresponding changes in text coherence and the overall success in promoting or suppressing the target themes.

### B.1. Deactivation of theme

In the deactivation task, we target a theme we call "Scientific concepts and entities," which is related to black holes, quantum mechanics, and other advanced physics topics. The full list of related features is presented in Table 2.

**Generation and evaluation setup.** We use the prompt, *"I think that the biggest problem of contemporary theoretical physics is "*, and generate text with a maximum length of 96 tokens, $\text{top}_p = 0.7$, and temperature $T = 1.27$. To determine whether each theme is present in the generated text, we query a gpt4o-mini language model for a score from 0 to 5 on each theme, following an approach similar to Chalnev et al. (2024). We use the following system prompt for validation:

```
"""You will receive a text that was generated by some language model.

You need to evaluate this text based on the following criteria:

1. Coherence score: text is gramatically correct, language is clear and coherent.
2. Behavioral score: text is directly related to some predefined subject.

The subjects are:
a) ** Personalities, their influence and achievements **. Text may contain
references to individual or group achievements or lack of results, discussion
about Nobel prizes etc.
```

| Theme | Feature index | Interpretation |
|---|---|---|
| Anger and frustration | 12/res/4111 | expressions of anger and frustration |
| Mental health issues | 12/res/16226 | ref. to mental health issues and their connections to other health conditions |
| Wedding and marriage | 12/res/4230 | terms related to weddings and marriage ceremonies |
| Religion and God | 12/res/5483 | spiritual themes related to faith and divine authority |

*Table 3.* Initial choice of feature for activation task.

```
b) ** Scientific concepts and entities **. Text relates to some specific
scientific concepts, such as particles, black holes, particular theories,
Standard Model, quantum wavefunctions, gravitation etc.
c) ** Research methodology and experimentation **. Text relates to methodology
and philosophy of science, scientific research techniques and procedures,
especially method of experiment, empirical basis of theories etc.

Rate the behavioral score in this order, i.e. [a, b, c].

Give an integer score from 0 to 5 for each criteria.
Do not judge text based on ethical considerations, preferences etc., it is
important for that evaluation to be objective.

Format your output as JSON:

{
    "coherence": 5,
    "behavioral": [1, 1, 1]
}

"""
```

And the following input formatting:

```
"""
Text:
\"\"\"
{text}
\"\"\"
"""
```

## B.2. Activation of theme

We select the features used in Chalnev et al. (2024) (Table 3), so we do not significantly alter the initial feature choices.

**Flow graph building.** Starting from the target feature, we build a flow graph forward and backward, computing similarity scores $s^{(R)}, s^{(M)}, s^{(A)}$ for each residual feature, referencing its predecessors. We then cut our graph on layers where $s^{(R)}$ is below a threshold value $t^{(R)} = 0.5$, forming the similarity span from $l_{\text{start}}$ to $l_{\text{end}}$. We also remove features from modules using a threshold value of 0.15.

**Feature activation transformation.** For steering, we add scaled decoder columns of selected features:

$$\mathbf{h}_t \leftarrow \mathbf{h}_t + \mathbf{s} \cdot \mathbf{V}^\mathsf{T},$$

where $\mathbf{s} \in \mathbb{R}^f$ is a vector of scaling coefficients for $f$ features whose embeddings are in $\mathbf{V} \in \mathbb{R}^{d \times f}$. We apply this transformation to all tokens to globally promote or suppress certain features.

**Distribution of steering coefficient.** To steer multiple related features, one can distribute a single steering coefficient across them, rather than manually tuning each feature. We consider two main strategies.

Let $s$ be the initial scaling coefficient and $l$ the layer index. We define **exponential scaling** of a related feature as:

$$s' = s \times e^{\alpha l},$$

and **linear scaling** as:

$$s' = k \times l + b, \quad \text{where} \quad k = \frac{s^* - s}{l_{\text{end}} - l_{\text{start}}} \quad \text{and} \quad b = s - k \times l_{\text{start}},$$

where $l_{\text{start}}, l_{\text{end}}$ are the first and last layers on the obtained graph.

Our intuition is that the more features we steer across different layers, the less strength each subsequent feature should have. Exponential scaling with $\alpha < 0$ causes $s'$ to decrease monotonically. Linear scaling is essentially an interpolation between $(l_{\text{start}}, s)$ and $(l_{\text{end}}, s^*)$. **Constant scaling** is a special case of both methods where $s' = s$.

We also employ folding (Balagansky et al., 2024) to incorporate information about typical activation levels of different features, helping us distribute the coefficient more effectively. We set $\alpha = -0.05$ and $s^* = 1$, based on generating a small batch of test completions and manually checking the trade-off between coherence and theme intensity.

The *single* method reported in 5.3 employs a per-layer activation strategy, and other methods are cumulative.

**Generation and evaluation setup.** We use the prompt, *"I think "*, a maximum generation length of 36 tokens, $\text{top}_p = 0.7$, and temperature $T = 1.27$. We validate the result with the following system prompt:

```
"""You will receive a text that was generated by some language model.

You need to evaluate this text based on the following criteria:

1. Coherence score: text is gramatically correct, language is clear and coherent.
2. Behavioral score: text has explainable relation to some predefined subject.

Give an integer score from 0 to 5 for each criteria.

Do not judge text based on ethical considerations, preferences etc., it is
important for that evaluation to be objective.

To evaluate behavioral score, also consider related terminology or entities
which may be not directly discuss the subject, but associated with it. The name
of the subject is just an orienting point for further search of ** explainable **
connection to that theme in text. For example, if subject is the History of Rome,
then strong presence of latin or italian language, or discussion about not
history, but architecture of ancient Rome should also lead to high behavioral
score, because it has strong explainable connection.

Format your output as JSON:

{
    "coherence": 5,
    "behavioral": 1
}

"""
```

And the following input formatting:

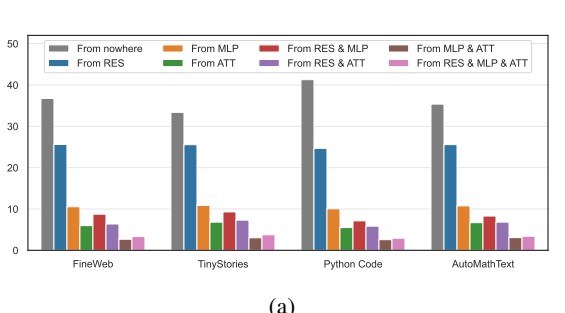

(a)

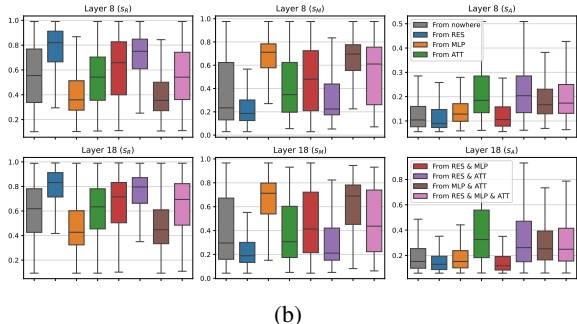

(b)

*Figure 12.* (a) Percentage of feature groups obtained for each dataset. (b) Distribution of scores for layers 8 and 18. We observe a clear distinction between groups, which additionally indicates the validity of the proposed method.

*Figure 13.* Probability of group A (row) to appear in group B (column), aggregated over all layers. For example, if we take the "From ATT" group, then with a probability of 0.45, features from this group would appear in the "From RES & ATT" group. High scores for the "From nowhere" group represent its stochasticity.

```
"""Subject: {theme}
Text:
\"\"\"
{text}
\"\"\"
"""
```

## C. Additional results for experiments

### C.1. Identification of feature predecessors

The "From nowhere" group is the most present among all other groups (Figure 12(a)). This may be the consequence of sporadic activation of some features or a matching error. The absence of groups with attention module is probably the consequence of our training procedure, which clearly contrasts with the distribution for Llama Scope (Figure 19(a)).

In Figure 12(b), we see that certain groups are indeed distinct with respect to corresponding similarity scores, which we describe in Section 5.1. Figure 14 shows the percentage of tests passed with a p-value threshold 0.001 for each pair of groups, aggregated for each layer and dataset.

However, we observe that features may fall into different groups depending on the context and chosen token (Figure 13), which indicates that we need to estimate, for every feature, the most probable groups they fall into.

A three-part partition of the group distributions for both Gemma Scope (Figure 5) and Llama Scope (Figure 19(b)) aligns with earlier observations on monosemanticity of neurons across layers (Gurnee et al., 2023). Partitioning the model into the first 20%, the next 40%, and the final 40% of layers reveals varying degrees of monosemanticity, which may have a connection with the three-part partition in the distribution of groups across layers – for Gemma Scope, we have mentioned

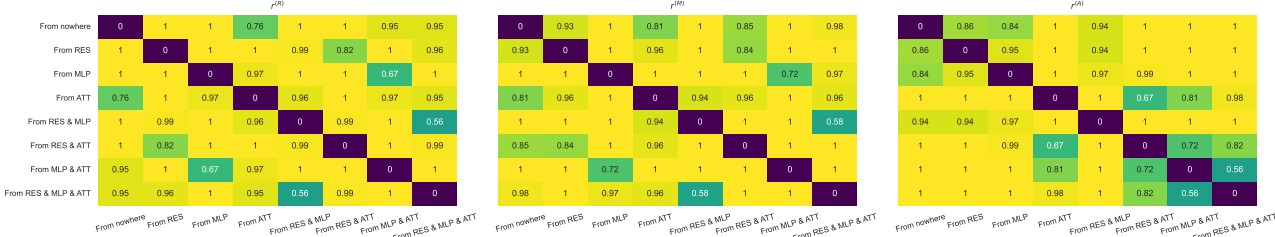

*Figure 14.* Percentage of statistically significant differences between groups with respect to a certain score.

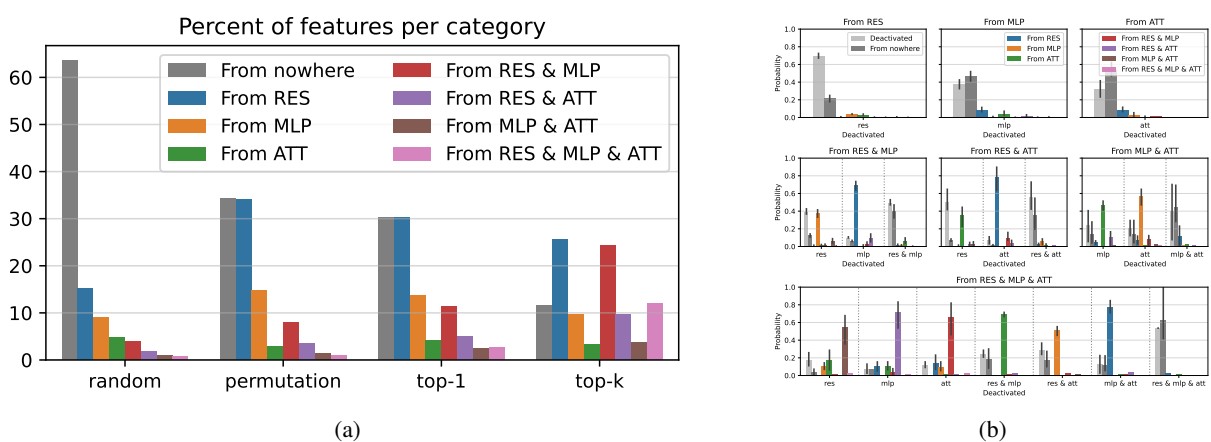

*Figure 15.* (a) Percentage of features per each method. There was a total of 13106 activated features, and for every feature, four matching strategies were applied. We see that $top_5$ method detects many more combined groups than other methods, especially "From RES & MLP". (b) Probability for a feature from some group $A$ (labeled as the subplot title) to become from group $B$ (shown in legend) after deactivation of some predecessor. Each bar shows the percentage of times the feature falls into a new category.

parts [0, 5], [6, 15], and [16, 25], and in the case of Llama Scope we observe segments [0,8], [9,16], and [17,31].

### C.2. Deactivation of features

We observe that the $top_5$ method happens to detect many more activated predecessors than other methods, and detects more combined groups as depicted in Figure 17(a).

Deactivation of a residual predecessor in the case of "From RES & MLP" and "From RES & ATT" with almost equal chance also deactivates the predecessor on the corresponding module or deactivates the target feature entirely, as depicted in Figure 15(b). This suggests that in those cases, the residual predecessor is indeed blocked from further propagation. However, in most cases, full deactivation (of all predecessors) is required to deactivate the target feature. In many cases, "Deactivated" and "From nowhere" are equally probable, which indicates remaining causal dependencies that we miss with our method.

We also observe the appearance of new groups, i.e., a feature might initially be "From MLP", but after deactivation of the MLP feature (which is actually a re-calculation of the full forward pass with intervention on the MLP module), we observe that sometimes the feature might have new predecessors, for example, on the attention module. This is unexpected since the MLP module actually comes after the attention module, but the presence of such groups is not so strong, so we think of it as sporadic behavior of internal computations.

We must take into account that there may be other causal relations for some feature to appear, for instance, interaction between different tokens on the attention module, different features, different modules, or even different layers. Furthermore, the appearance of a feature means a certain structure of the hidden state, and this structure was built by many previous layers where information was somehow encoded in a complex way by the interaction of many different components and

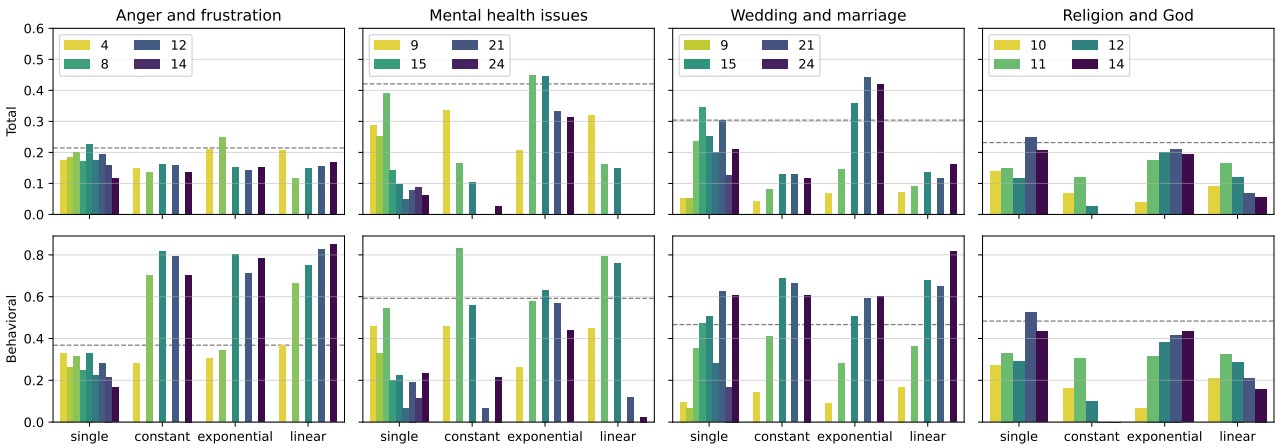

*Figure 16.* From each flow graph, we select features on a particular layer $l$ and perform steering with the four different strategies. Bars represent the best result for each layer among all scores $s$. In some cases, steering on a layer other than 12 may improve results.

features. This is like an optimization process in a non-convex scenario, which may converge to some local minima with certain properties, and the information processing inside the model may converge to a certain structure of hidden states with certain semantics contained in it.

Therefore, in an ideal situation, to really deactivate some feature, we must somehow influence the hidden state to behave as if there *never* had been such a feature, its evolutionary ancestors, or any other causal predecessors, and they had never been involved in information processing. Our current steering procedure works in a "neighborhood" of some local hidden state "minima", but efficient deactivation consists of changing the convergence direction toward another hidden state "minima" at an early stage of computations. This most likely also applies to the activation of some feature.

### C.3. Model steering

We also measure the effect of steering on different layers. The best result among all available $s$ is shown in Figure 16. Note that *single* is a per-layer method, while the others are cumulative. We see that different layers perform differently, and while the initial features were located at the 12th layer, sometimes the best layer is located elsewhere.

We also have performed a small experiment to test the activation of another theme with many flow graphs using the same prompt as in the deactivation case. We start with the features described in Table 4 and build flow graphs from them. Then we manually choose some of the subgraphs based on semantic considerations and threshold values. The total amount of features selected on different layers is presented in Figure 15(a). After that, we steer the resulting features with manually obtained $s = 8$ and $\alpha = -0.05$ for the single-layer case, and $s = 3$ and $\alpha = -0.25$ with the exponential decrease method for the cumulative setting.

By using our method, we found influential features on the 5th layer that gave us the best result among all other layers (Figure 17(b)), while none of the initially found features were placed on that layer. However, we did not tune the hyperparameters properly, so there may be room for another conclusion.

### C.4. Comparison with Pearson Correlation Baseline

While data-driven methods provide useful insights, they face challenges with sparse SAEs and low-frequency features. Our data-free approach overcomes these limitations through adjustable top-k matching, particularly advantageous in sparse activation regimes.

We evaluated Pearson correlations on 100K non-special tokens from FineWeb's "default" subset for features in Gemma Scope's even layers and all layers of Pythia-70M-Deduped and GPT-2. Using an expanded sample size of 500 (Appendix A), we identified feature groups. Figure 18 presents results for the Gemma model.

| Feature index | Interpretation from Neuronpedia |
|---|---|
| 12/res/6778 | references to testing and experimentation processes |
| 16/res/13806 | references to experimental studies and methodologies |
| 18/res/1056 | references to experiments and experimental protocols |
| 18/res/7505 | terms and phrases related to research activities and methodologies |
| 23/res/10746 | terms related to modeling and model-building in scientific contexts |
| 24/res/11794 | terms and phrases related to scientific reasoning and methodology |
| 24/res/1027 | concerns related to study validity and bias in research methodologies |
| 24/res/7391 | phrases related to inquiry and questioning |
| 24/res/1714 | references to academic studies and their outcomes |
| 25/res/6821 | terms related to experimental methods and results in scientific research |

*Table 4.* Features initially chosen for activation of "Research methodology and experimentation" theme.

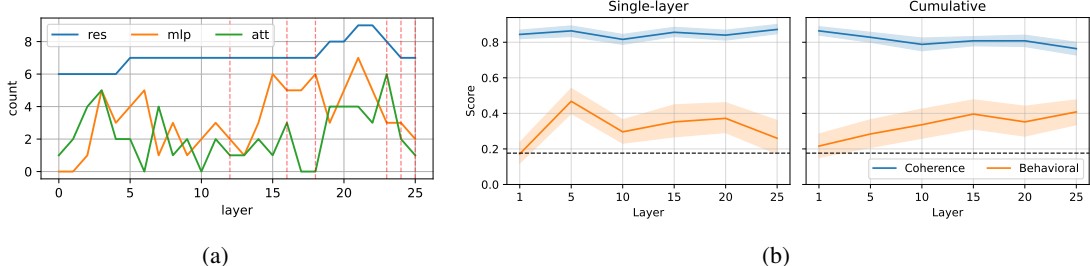

(a)                (b)

*Figure 17.* (a) Amount of features selected for activation of "Research methodology and experimentation" theme. Vertical lines represent the placement of the initially selected features. (b) Results for steering of selected features. Score is a total metric measured as Behavioral × Cumulative. We can see that despite none of the initial features being placed on the 5th layer, it gives us the best result.

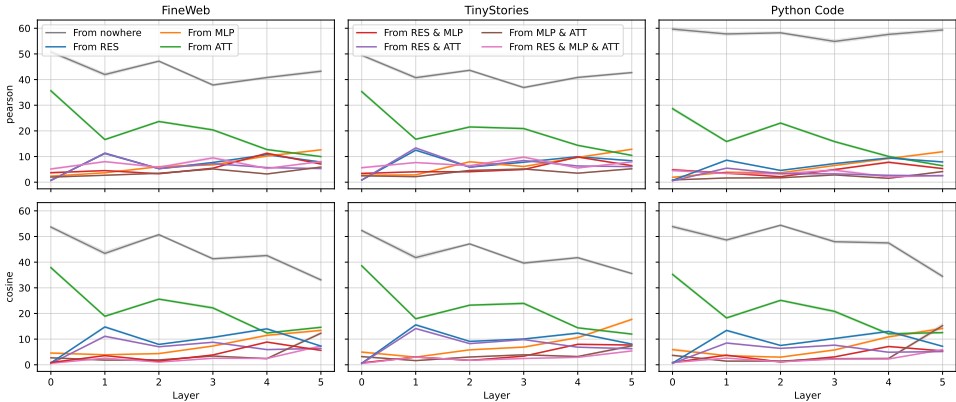

*Figure 18.* Feature group identification comparison (Section 5.1) between $\text{top}_1$ cosine similarity and Pearson correlation. While correlation better captures predecessors with under-trained embeddings, it exhibits stronger dataset dependence and sparsity sensitivity.

Correlation-based matching reduced the "From nowhere" group presence and improves predecessor identification on attention module, though potential misalignment between our attention SAEs and Gemma Scope's residual/MLP SAEs may affect quality. Results aligned closely with Llama Scope for Pythia (showing clearer attention features), while GPT-2 displayed increasing "From nowhere" presence in later layers.

The correlation method failed to consistently outperform $\text{top}_1$ cosine similarity, particularly on out-of-distribution Python code. Strong agreement emerged between methods for Gemma Scope and GPT-2 residual SAEs, but weaker alignment occurred for module-based SAEs, consistent with prior feature propagation studies.

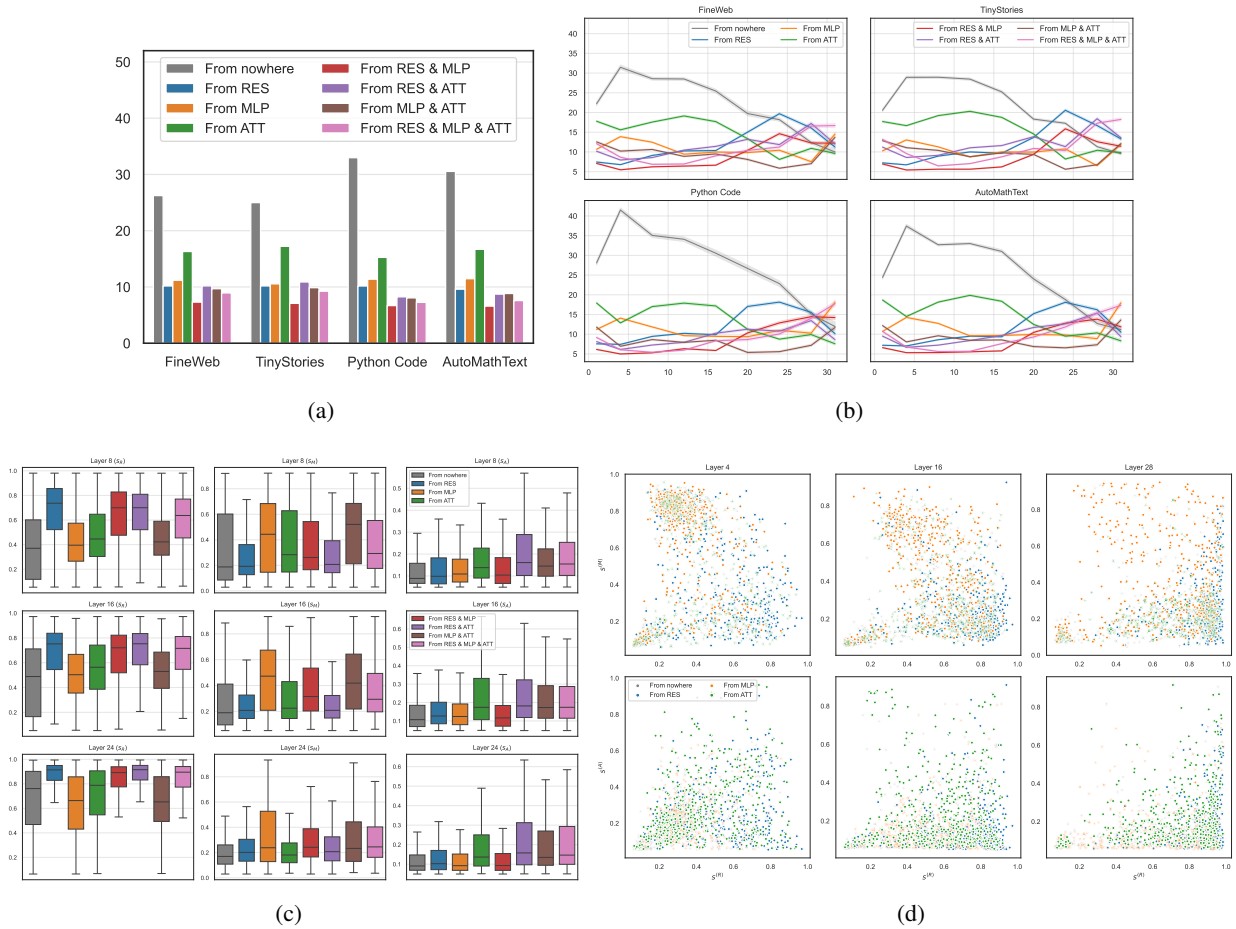

(a)

(b)

(c)

(d)

*Figure 19.* (a) Distribution of groups for Llama Scope. We observe a clear distinction from Gemma Scope results (Figure 12(a)) due to a much smoother distribution. This may be a consequence of various factors: the architecture of the models or SAEs, the training procedure, differences in data distribution, etc. (b) Distribution of groups across multiple layers. We observe approximately the same pattern as for Gemma Scope (Figure 5), indicating shared properties between the models. (c) Distribution of scores for different groups. We see that the groups are slightly less distinct from each other compared to the case of Gemma Scope (Figure 12(b)), but they are still present. This is also reflected in (d) the separability of different groups based on their cosine similarity relations.

## D. Experiments with Llama Scope

We have also used the Llama Scope SAE pack (He et al., 2024) to evaluate our proposed approach and have found that it is well aligned with the results we observe for Gemma Scope. However, we did not perform a steering experiment or graph building, and we consider it as one of the future study directions. For these SAEs, the main picture remains the same.

First, they have a more uniform distribution of feature groups, with a clear prevalence of attention features (Figure 19(a)). This indicates that our attention SAEs for Gemma were perhaps not trained well enough. We suspect that experiments with other models will show that Llama Scope results are more accurate with respect to predecessors distribution.

Second, despite the uniformity, we observe that Llama Scope groups are slightly harder to separate from each other in terms of similarity scores (Figures 19(c) and 19(d)), which may also be the consequence of the different architecture of the model itself or because these SAEs initially were trained with the TopK activation function.

Third, the dynamics of the group distribution is slightly different (Figure 19(b)), but the overall pattern (with a three-part separation and an increase of "From RES" in the latter layers) and overall percentage is still approximately the same. Perhaps we may interpret this similarity between Llama Scope and Gemma Scope as an argument for the validity of our analysis; however, it still requires experimentation with other architectures and sizes.

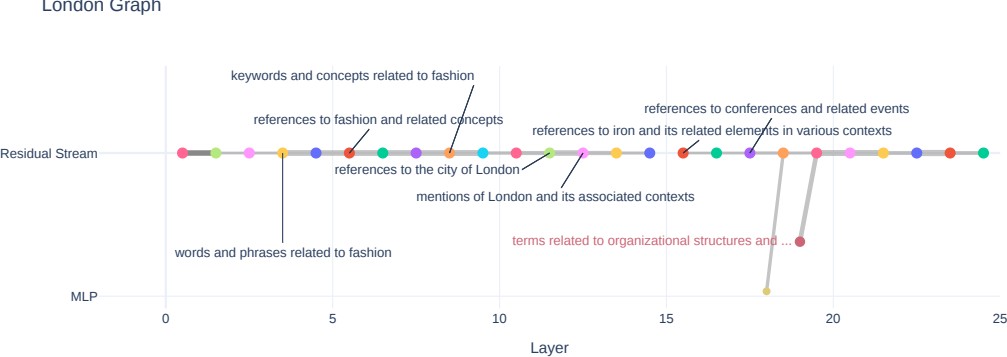

*Figure 20.* Flow graph for the 12/res/14455 feature. As reported in Chalnev et al. (2024), steering of that feature might produce themes related to fashion, and we clearly observe that our flow graph captures this semantics in the earlier layers.

## E. Examples of flow graphs

In this section, we describe some of the interesting flow graphs we have found. For simplicity, we denote each feature as "layer index / module / feature index".

**Particle physics graph.** We start with the graph presented in Figure 2 that was built from feature 24/res/14548. Once we obtain it, we might explore how its semantics evolved across different layers. The full list of features with interpretations that belong to this graph is presented in Table 5.

From the first to the sixth layers, we have semantics mainly related to experiments and abstract particle physics. Then we have feature 7/res/16335 with the following scores: $s^{(M)} = 0.82$ for 7/mlp/6110, with semantics related to datasets and measurements, and $s^{(R)} = 0.3$ for 6/res/2452 with semantics about Dark Matter. After this, the semantic flow has a tighter connection to measurement and observation-related themes, while maintaining the quantum physics semantics.

We hypothesize the following relation: initially, the flow graph was related to science and experiment, and on the 7th layer it was transformed in a way that 7/mlp/6110 introduced a slightly new semantics to the already existing one, perhaps also replacing the vague "experimentation" theme. Thus, we think of this interaction as an example of a linear circuit, and feature 7/res/16335 falls into the *processed* category.

After the 7th layer, we observe a slight strengthening of particle physics semantics, perhaps because of some other interactions, while also introducing the bosons theme. From this layer, $s^{(R)}$ is large and $s^{(M)}$ is small. On the 17th layer, we encounter feature 17/res/8130 with $s^{(R)} = 0.48$ and $s^{(M)} = 0.79$ for 16/res/10649 and 17/mlp/8454, respectively. The MLP feature relates to gauge theories and theoretical matters, and the feature 17/res/8130 drifts toward gauge bosons and their interaction theme. We also hypothesize that at this particular point, the feature from MLP added new information to the already existing one; therefore, 17/res/8130 is also a *processed* feature.

After this, the semantic meaning sticks more to the Standard Model and particle interaction, but with less practical (such as measurements and data) and more theoretical aspects. We can also see that MLP features on layers from 20 to 24 are more related to theoretical aspects than their residual matches.

**London graph.** An interesting observation was made in Chalnev et al. (2024): steering feature 12/res/14455 with interpretation "mentions of London and its associated contexts" with a large steering coefficient led to the generation of a theme related to fashion, design, and exhibition. If we build the flow graph from this feature, we observe that in the first half it clearly has fashion-related semantics (Figure 20). This indicates that feature 12/res/14455 contains the semantics of its evolutionary predecessors. We also see feature 17/res/9260 with references to conferences (followed by feature 18/res/2010 with the same semantics), which relates to shows and exhibitions mentioned in Chalnev et al. (2024). Perhaps we might interpret this particular flow graph as "references to fashion and design exhibitions performed in London".

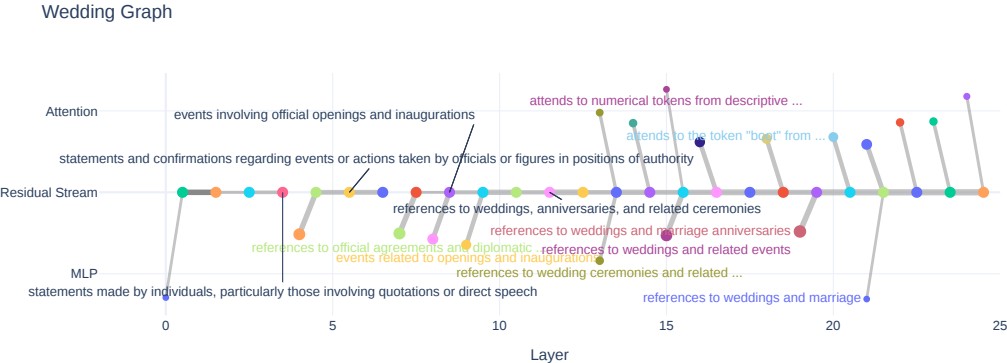

*Figure 21.* Flow graph for the 12/res/4230 feature. In this case, we observe that the second half of the model is closely related to wedding and marriage ceremonies. We believe that the "official" aspect in the interpretation of features in earlier layers is closely related to the fact that wedding ceremonies and marriage are themselves official procedures—the registration of a specific type of interpersonal relationship.

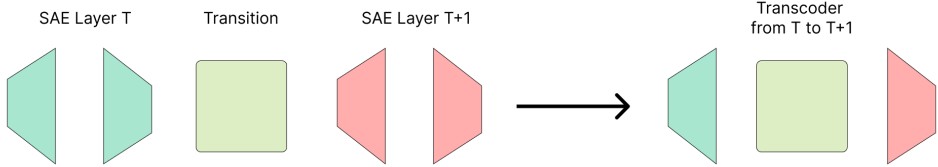

*Figure 22.* Two SAEs with a learned transition matrix $T$ can be seen as a transcoder from layer $t$ to layer $t + 1$.

**Wedding and marriage graph.** We have observed in our experiments that steering feature 12/res/4230 with interpretation "terms related to weddings and marriage ceremonies" indeed increases the presence of ceremony-related tokens in a wedding context. If we obtain a flow graph for that feature, we see that it begins with themes related to official meetings and agreements, suggesting that the "ceremony" part of the flow graph interpretation may arise from this official context.

We conclude that these flow graphs may be used not only for interpretation and understanding of feature evolution, but they can also explain the outcomes of certain steering procedures.

## F. Similarity between Matching and Transcoders

Dunefsky et al. (2024) proposed using transcoders to study computational graphs, and Balagansky et al. (2024) proposed using a permutation $\mathbf{P}$ to find matching features across layers. In this section, we study the similarity between these two methods. Similarly to Balagansky et al. (2024), we chose explained variance as a metric to measure the quality of the translayer transcoder. See Figure 22 for the schematic overview of the transcoders obtained by transition mapping $\mathbf{T}^{t \rightarrow t+1}$.

**Setup.** We use SAE trained on the residual stream after layers 14 and 15. We vary the methodology to find and apply the transition $\mathbf{T}$. In our cases, we consider only a linear map so that $\mathbf{T} \in \mathbb{R}^{|\mathcal{F}| \times |\mathcal{F}|}$.

First, we study how folding proposed in Balagansky et al. (2024) affects final transition performance. Results are presented in Figure 23. From these results, we conclude that folding is useful in the inference case to match different scales of activations across layers; in contrast, it has almost no effect on finding permutations, with the exceptional case of incorporating $b_{enc}$ to find permutations. Notably, the baseline with a simple approach of finding cosine similarity outperforms permutations.

Second, we compare cosine similarity with other methods to obtain the transition map $\mathbf{T}$. Instead of relying on a matrix containing 0 and 1, we use the $\text{top}_k$ operator. Results are presented in Figure 24. Interestingly, folded $\text{top}_2 \boldsymbol{W}_{dec}^{(14)\top} \boldsymbol{W}_{dec}^{(15)}$ outperforms the permutation baseline; however, cosine similarity ($\mathbf{I}_{x>0} \text{top}_1 \boldsymbol{W}_{\text{dec}}^{(14)\top} \boldsymbol{W}_{\text{dec}}^{(15)}$) performs best.

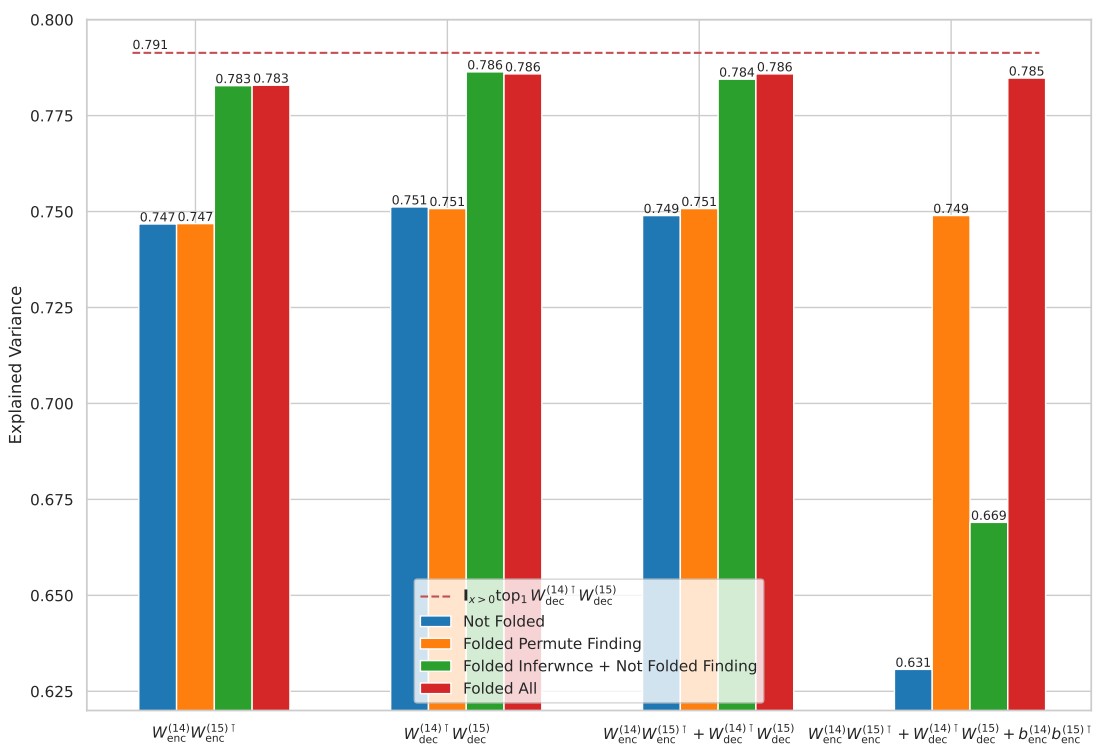

*Figure 23.* Explained variance of the various permutation variants. ($\mathbf{I}_{x>0} \, \text{top}_{1} \boldsymbol{W}_{\text{dec}}^{(14)\top} \boldsymbol{W}_{\text{dec}}^{(15)}$) performs best. See Appendix F for more details.

Cosine similarity between decoders' vectors

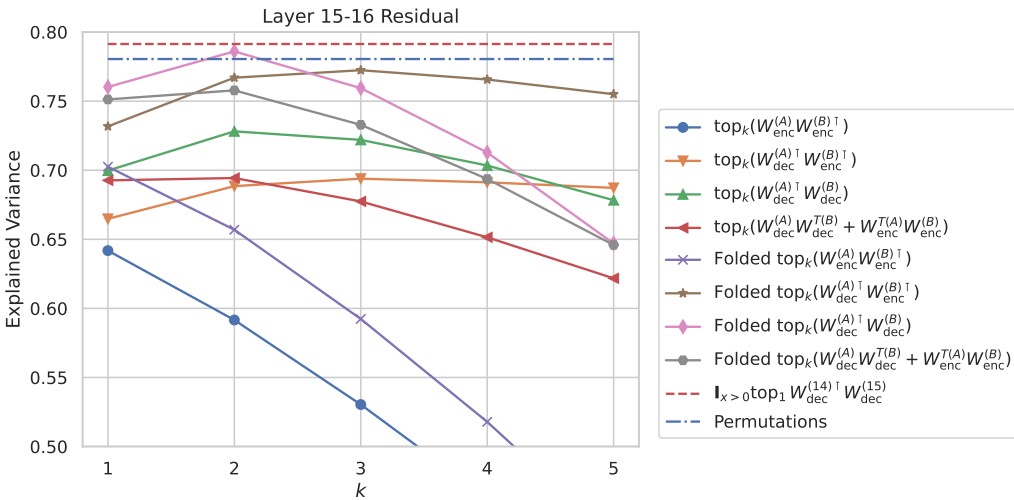

*Figure 24.* Comparison of various $k$ in $\text{top}_k$ operator and different weights of the SAE. Cosine similarity ($\mathbf{I}_{x>0} \, \text{top}_{1} \boldsymbol{W}_{\text{dec}}^{(14)\top} \boldsymbol{W}_{\text{dec}}^{(15)}$) performs best. See Appendix F for more details.

| Layer | Feature index | Interpretation |
|---|---|---|
| 0 | 0/mlp/12987 | punctuation, particularly quotation marks and dialogue indicators |
|   | 0/res/14403 | elements that indicate neglect or care in familial relationships |
| 1 | 1/mlp/16168 | mentions of astronomical phenomena and their characteristics |
|   | 1/res/13755 | metaphorical language and scientific terminologies related to variables and coefficients |
|   | 2/res/12939 | numerical data or metrics related to surveys and observations |
|   | 3/res/16138 | scientific terminology related to study results and causes |
|   | 4/res/11935 | terms related to particle physics and their interactions |
|   | 5/res/14558 | numeric or symbolic representations related to mathematical notation or scientific data |
|   | 6/res/2452 | key terms related to Dark Matter detection and experimental setups |
| 7 | 7/mlp/6110 | terms related to datasets and classification in statistical or machine learning contexts |
|   | 7/res/16335 | technical terminologies related to particle physics measurements |
|   | 8/res/9666 | scientific measurements and data related to particle physics |
|   | 9/res/8318 | references to particle physics concepts and measurements |
|   | 10/res/13754 | technical terms and measurements related to particle physics |
|   | 11/res/7614 | terms related to particle physics and specifically the properties of W and Z bosons |
|   | 12/res/2812 | statistical terms and measurements associated with quark interactions |
|   | 13/res/4955 | terms and concepts related to particle physics experiments and measurements... |
|   | 14/res/5262 | keywords related to particle physics, specifically concerning quarks and their properties |
|   | 15/res/9388 | concepts related to particle physics measurements and events |
|   | 16/res/10649 | complex scientific terms and metrics related to particle physics experiments |
| 17 | 17/mlp/8454 | theoretical concepts and key terms related to physics and gauge theories |
|   | 17/res/8130 | terms related to gauge bosons and their interactions within the context of particle physics |
|   | 18/res/11987 | technical and scientific terminology related to particle physics |
|   | 19/res/15694 | references to scientific measurements and results related to particle physics... |
| 20 | 20/mlp/601 | terms associated with quantum mechanics and transformations |
|   | 20/res/12523 | terms and concepts related to particle physics and the Standard Model |
| 21 | 21/mlp/594 | technical terminology and classifications related to data or performance metrics |
|   | 21/res/14511 | scientific terminology and concepts related to fundamental physics... |
| 22 | 22/mlp/14728 | references to gauge symmetries in theoretical physics |
|   | 22/res/11460 | terms and concepts related to particle physics and theoretical frameworks |
| 23 | 23/mlp/6936 | terms related to theoretical physics and particle interactions |
|   | 23/res/9592 | terms related to particle physics and their interactions |
| 24 | 24/mlp/11342 | terms and concepts related to theoretical physics and particle interactions |
|   | 24/res/14548 | terms and references related to particle physics and standard model parameters |
|   | 25/res/1646 | technical terms and measurements related to particle physics and the Standard Model |

Table 5. Graph built from 24/res/14548 feature with MLP features dropped by threshold $t^{(M)} = 0.25$.

