# OpenReview forum: "Analyze Feature Flow to Enhance Interpretation and Steering in Language Models"
_ICML.cc/2025/Conference — ICML 2025 poster_

### Official Review · Reviewer_t2gC · 2025-03-11

**Overall Recommendation:** 3

**Summary:**

This paper studies sparse autoencoders (SAEs) trained on different layers and modules (residual stream, MLP and attention), proposes using cosine similarity to locate predecessor features in the previous layer given any target feature represented by SAE decoder weights. Through this approach, this paper traces how SAE features evolve throughout the model, and constructs a flow graph that connects features in different layers that allows for effective multi-layer cumulative steering for text generation.

**Claims And Evidence:**

Yes. However, I have the following concerns regarding certain parts of the paper:
- **Section 5.1, Identification of feature predecessors.** The authors illustrate the (statistical) difference between each group of features, categorized by how each feature is co-activated with its predecessors. From Figure 5, it is noted that among the target features examined, **about 60-70%** can be explained as originated from RES or being “created” via MLP/Att. However, I have concerns about this estimand as follows.
    - (1) The target feature set is identified through random sampling (Appendix A.1), which might include features that are generally activated but not specific to the dataset considered. To further validate the goal in Section 3.3 of “tracking the evolution of feature”, it would be nice to improve this step by ensuring the features considered are rarely activated on other datasets so they are tailored to the examined distribution. This might also explain why the curves look similar to each other albeit obtained from drastically different datasets.
    - (2) It is unclear whether there are preceding features that are co-activated with the target feature with high probability, **but cannot be identified via cosine similarity search.** Thus, it is necessary to examine all preceding features to identify candidates with the highest frequency of co-occurrence with the target feature, and verify if these features truly correspond to the ones determined by the cosine similarity search.
- **Section 5.2, Deactivation of features.** My concern is whether a cascading effect of deactivation could exist, given the discussion in Section 2.3 that “most features in the residual stream remain relatively unchanged across layers”. Specifically, is it possible that a similar feature is reactivated again in later layers, even though it appears to be eliminated when examining the current target feature? My suggestion is to check the downstream features in the constructed flow graph, to determine if such an event occurs.
- **Section 5.3, Model steering.** From Figure 10, how could the conclusion that “cumulative intervention outperforms the single-layer approach” be drawn, given that the blue curve shows a better best score than the orange curve?

**Essential References Not Discussed:**

I have not found essential missing references.

**Experimental Designs Or Analyses:**

Yes. For questions, see claims and evidence.

**Methods And Evaluation Criteria:**

Yes. The proposed use of cosine similarity has already been adopted for clustering SAE features (e.g., [1]), and in this paper it is taken to examine feature evolution across different layers. The evaluation of deactivation and steering follows the intervention standard of SAEs in current literature.

[1] Scaling Monosemanticity: Extracting Interpretable Features from Claude 3 Sonnet

**Other Comments Or Suggestions:**

- Section 2.2 line 055: wrong reference to transcoder
- Section 3.2 line 144: should the topK operation be executed per row, instead of been applied globally?
- Caption in Figure 10: the blue curve is not a multi-layer steering method.

In addition, I strongly recommend that the authors move some essential information from Appendices A and B into the main text. The authors emphasize from the beginning that the technique is “data-free”, without mentioning how the features are selected given a dataset of interest in the main text. Additionally, there are technical details omitted in the main text that are required to understand the experimental results. For example, the labels of “random” and “permutations” and the $r$ parameter in the deactivation experiment and the “cumulative” terminology. For my other concerns, please refer to claims and evidence.

**Other Strengths And Weaknesses:**

The evaluation is comprehensive, and the idea of tracing the evolution of features through SAEs is novel to date.

Though cumulative intervention is claimed to be beneficial, I don’t see the effect significantly, see claims and evidence.

**Questions For Authors:**

Please see claims and evidence.

**Relation To Broader Scientific Literature:**

This paper provides evidence on how SAE features evolve through different layers. The enabled automatic similar feature search (flow graph) and the following intervention proposal could enhance the quality of model steering and contribute to a better understanding of these models.

**Theoretical Claims:**

N/A

---

> ### Author Rebuttal · Authors · 2025-04-01
>
> Thank you for valuable questions.
>
> > To further validate the goal in Section 3.3 of “tracking the evolution of feature”, it would be nice to improve this step by ensuring the features considered are rarely activated on other datasets so they are tailored to the examined distribution.
> >
>
> We appreciate this suggestion. To address feature specificity, we estimated activation frequencies across 100k non-special tokens from FineWeb and categorized features into quantiles based on their activation rates. As expected, the results vary significantly depending on the quantile selected, supporting the need for careful feature selection.
>
> https://anonymous.4open.science/r/icml_rebuttal-5064/quantiles.png
>
> > It is unclear whether there are preceding features that are co-activated with the target feature with high probability, **but cannot be identified via cosine similarity search.**
> >
>
> To evaluate this, we calculated the following score: the fraction of features for which the top-1 match by cosine similarity also appears in the top-*k* matches by Pearson correlation.
>
> https://anonymous.4open.science/r/icml_rebuttal-5064/top_k_corr.png
>
> https://anonymous.4open.science/r/icml_rebuttal-5064/top_k_gpt.png
>
> https://anonymous.4open.science/r/icml_rebuttal-5064/top_k_pythia.png
>
> We observe strong agreement for residual features in Gemma and GPT-2, while MLP and attention features show lower consistency. However, we have not yet validated whether Pearson-based matches reliably predict causal relationships or steering outcomes—a valuable direction for future work, and we thank you for highlighting this.
>
> > is it possible that a similar feature is reactivated again in later layers, even though it appears to be eliminated when examining the current target feature?
> >
>
> Yes, this is very common. Since deactivation typically applies only to a small subset of features, and rescaling coefficients are often modest, hidden states undergo only minor perturbations, allowing for later reactivation. We hypothesize this stems from the model’s self-repair mechanisms, which can recover information even after layer pruning (cf. [1–3]). To mitigate this, effective deactivation may require not just zeroing target features but also adding another steering vectors to the hidden state to prevent such recovery (discussed in Appendix C.2).
>
> To test the occurrence of reactivation, we used three strategies: deactivating only the first layer in a graph, deactivating a random layer in a graph, deactivating the first half of layers in a graph. Subsequent deactivations were measured on residual nodes that comes after the deactivated layer, confirming that reactivation is present, but it depends on intervention method and strength.
>
> https://anonymous.4open.science/r/icml_rebuttal-5064/reactivation.png
>
> > From Figure 10, how could the conclusion that “cumulative intervention outperforms the single-layer approach” be drawn, given that the blue curve shows a better best score than the orange curve?
> >
>
> Thank you for highlighting this. Cumulative steering generally achieves comparable effects at *lower rescaling coefficients* (Figure 10), improving stability. While single-layer steering can yield higher total scores, it requires either precise selection of highly impactful layers, or spanning multiple layers with a well-constructed feature graph.
>
> This introduces additional optimization challenges not faced by cumulative steering. We will revise this conclusion in the updated manuscript to reflect this nuance.
>
> References:
>
> [1] [Your Transformer is Secretly Linear] (https://arxiv.org/abs/2405.12250)
>
> [2] [What Matters in Transformers? Not All Attention is Needed] (https://arxiv.org/abs/2406.15786)
>
> [3] [The Hydra Effect: Emergent Self-repair in Language Model Computations] (https://arxiv.org/abs/2307.15771)

---

> > ### Comment · Reviewer_t2gC · 2025-04-04
> >
> > Thanks for answering my questions.
> >
> > **Feature specificity.** You stated the frequencies are estimated using 100K FineWeb tokens. If I understand this correctly (please clarify this if I am wrong), you use **features sampled from the datasets you listed in the caption of the Y-axis** and test them on FineWeb tokens to produce the plots. If this is the case, I regard these results as indicating the sampled features in the paper **not dataset-specific**, since on TinyStories and Python Code there are a significant fraction of features that are also activated **frequently** on the FineWeb distribution. I raised this concern because, in Figure 5, the general trend across datasets is consistent, which can be due to the bias of using commonly activated but not dataset-specific features - when examining dataset-specific features (the new plots with lower quantiles), the dynamic change significantly (e.g., for the group of "From RES", it tend to decrease instead of increase). Thus, if the results across the paper are based on the full feature set, the difference between commonly-activated features and dataset-specific features is worth mentioning, because Figure 5 (and subsequent results) only represent results on the first group but not the second (as supported in the new plots). I would strongly encourage the authors to include new results on dataset-specific features, to check if there are other differences in subsequent experiments besides what you have for Figure 5; but I believe requesting these results is beyond the rebuttal period limit, so please take this as a suggestion.
> >
> > **Preceding features that are not selected through cosine similarity.** I raised this concern because the feature set discovered through cosine similarity can be incomplete, and the added results support this claim, especially for non-From-RES features. This may explain the phenomenon in Section 5.2 where negative rescaling has a strong impact on the From RES features but not MLP/Att ones.
> >
> > **Reactivation.** I regard this as a partial limitation of the work because this implies the feature flow discovered is incomplete, since deactivating a single (or several) feature does not fully cancel the resulting feature in the end. Given this work as the first to study SAEs for multi-layer understanding and steering, I suggest you add this to the discussion.
> >
> > ---
> > **Edit:** Thank you for the additional response to my concerns. Let me clarify my question on feature specificity. I understand in Figure 5 all results are derived independently from the datasets specified in the plot titles. However, this cannot guarantee that these features are **dataset-specific** if not filtered (as evidenced in your new experiments during rebuttal). This implies the same trend in Figure 5 could arise due to these commonly activated features. In short, the selection process dedicated to each dataset does *not* imply the set of features is also tailored to the data, as commonly activated features could exist. This is why I suggested checking the results of new experiments and examining when filtering is applied if the dynamic could be different - and it turns out to be true.
> >
> > Thanks for the additional comparison with the Pearson-based selection. This additionally highlights the data-free proposal. I also understand that the flow captured can be partial. Thanks again for addressing my concerns. I will raise my score.

---

> > > ### Author Response · Authors · 2025-04-06
> > >
> > > Thank you for your response.
> > >
> > > Let us clarify on points you mentioned above.
> > >
> > > **Feature specificity.**
> > >
> > > To clarify, the features validated in our paper (Figure 5) are derived independently from the datasets specified in the plot titles. For instance, in the Python code analysis, we perform a forward pass on the Python code dataset, collect information about activated features and their groups, and present the results on corresponding graph. This process involves a single pass over the dataset, ensuring that features are calculated independently for each dataset and are not pre-selected based on other datasets. Thus, the features we validate are inherently data-specific.
> > >
> > > The frequency plots (available [here](https://anonymous.4open.science/r/icml_rebuttal-5064/quantiles.png)) were constructed as follows: first, we estimate activation frequencies for all features on the FineWeb dataset and build a sets of features within each quantile, denoted as $F_q$. Next, we process the chosen datasets as described above to identify sets of activated features and their groups, denoted as $F_D$, where $D$ is the dataset name. To build the graph, we take intersection between quantile features and dataset features using $F_q \cap F_{D}$. This ensures that for TinyStories and Python Code, we only consider features rarely activated on FineWeb, addressing your requirement: "ensuring the features considered are rarely activated on other datasets."
> > >
> > > To fully address your feedback, we will include these results in Appendix C.1, along with graphs using frequencies computed on other datasets. We will also add deactivation experiment on other datasets to Appendix C.2 and reference them in Section 5.2.
> > >
> > > **Preceding features that are not selected through cosine similarity.**
> > >
> > > We acknowledge that feature interactions identified via cosine similarity may be incomplete, particularly due to inherent reconstruction errors in SAEs. To approximate dependencies, we use a simple heuristic: *“if a feature with a similar embedding activates in a prior module, it is considered dependent.”* While data-dependent Jacobian methods [1, 2] could enhance interaction estimation, we argue that tracing interactions through SAE weights alone remains valuable for practical applications like steering.
> > >
> > > Although MLP and attention layers yield fewer features than the residual stream, their explanations remain relevant (Figure 2), indicating meaningful contributions to the model’s behavior despite their smaller feature count.
> > >
> > > To assess how close our method is to top performance, we conducted an additional experiment comparing top-1 cosine similarity matching, top-1 Pearson correlation matching, and a full search for maximum achievable performance. The search process deactivates features one-by-one from the residual, MLP, and attention modules, and activation change metric was computed only for target features which group was identified by cosine or Pearson as either From RES, From MLP or From ATT to ensure fairness. With 1,894 features across two layers, each feature deactivated with three methods, we obtain:
> > >
> > > | Method | Top-1 Cosine | Top-1 Pearson | Search |
> > > | --- | --- | --- | --- |
> > > | Mean Activation Change | 0.75 | 0.74 | 0.83 |
> > > | Deactivated Features | 65% | 65% | 73% |
> > >
> > > The results show that Pearson correlation does not significantly improve upon cosine similarity, as noted in our response to reviewer jyW6, and highlight the value of our data-free strategy for both matching quality and causal analysis. We will include these results, computed on an extended feature set, in Appendix C.2.
> > >
> > > **Reactivation.**
> > >
> > > Consider a feature $r_i^l$ in the residual stream at layer $l$, which depends on both a feature $m_j^{l-1}$ from the previous MLP layer and a feature $r_k^{l-1}$ from the residual stream at layer $l-1$. Even if $r_k^{l-1}$ is deactivated, $r_i^l$ may still be activated via the alternative path through $m_j^{l-1}$. This illustrates our broader claim: features can be activated through multiple redundant pathways. In our reactivation experiments, we intentionally isolate and test the effect of deactivating one path at a time, acknowledging that this approach does not fully suppress the feature’s activation (due to remaining pathways).
> > >
> > > This underscores the complexity of feature interactions and highlights the value of our method in identifying partial dependencies—even if complete deactivation would require targeting all contributing paths. See also discussion in Appendix C.2 about maximum deactivation quality.
> > >
> > > Thank you again for your openness to further conversation. If we have addressed your concerns, we would greatly appreciate your reconsideration of our score.
> > >
> > > References:
> > >
> > > [1] Transcoders Find Interpretable LLM Feature Circuits. https://arxiv.org/pdf/2406.11944
> > >
> > > [2] Circuit Tracing: Revealing Computational Graphs in Language Models https://transformer-circuits.pub/2025/attribution-graphs/methods.html

---

### Official Review · Reviewer_vBgp · 2025-03-14

**Overall Recommendation:** 4

**Summary:**

The paper introduces a new approach to systematically map features discovered by SAEs across consecutive layers of LLMs. By using a data-free cosine similarity technique, the authors trace how specific features persist, transform, or first appear at each stage. This method yields granular flow graphs of feature evolution, enabling fine-grained interpretability and mechanistic insights into model computations.

The authors demonstrate how these cross-layer feature maps facilitate direct steering of model behavior by amplifying or suppressing chosen features, achieving targeted thematic control in text generation. Key contributions are threefold:
- Cross-Layer Feature Evolution: Using pretrained SAEs that can isolate interpretable monosemantic directions, the authors utilize information obtained from cosine similarity between their decoder weights to track how these directions evolve or appear across layers.
- Mechanistic Properties of Flow Graph: By building a flow graph, the authors uncover an evolutionary pathway, which is also an internal circuit-like computational pathway, where MLP and attention modules introduce new features to existing ones or change them.
- Multi-Layer Model Steering: The authors show that flow graphs can improve the quality of model steering by targeting multiple SAE features at once, and also offer a better understanding of the steering outcome.

**Claims And Evidence:**

From my perspective, the following are claims made by the authors (and their evidence), with my personal reasons to mark them as convinced or not:

1. **The authors believe that using a data-free cosine similarity technique can trace how specific features persist, transform, or first appear at each stage.**

    Evidence: some previous (Not really convinced)_

    From a **theoretical** perspective, this seems reasonable, as previous research has claimed high cosine similarity in nearby layers [1] [2]. Since how polysemantic hidden states are inherited with high similarity, it is reasonable to claim that the linear feature, as its components, are inherited to the next layers. However, from an experimental perspective, _I think the authors should give more illustrations of the cosine similarity results and analysis of features that are mapped to be the same.

2. **The four evolutionary pathways with cosine similarity.**

    (Relatively convinced) The clarification of four evolution paths, i.e., translated, processed, newborn, and not related, does in some way make sense. _However, I am curious about the activation patterns. I think the co-existence of activation values or the transference can be further evidence of these claims. e.g., if the feature of "Paris" is activated in two layers with the position predicted with the cosine similarity, this will verify the conclusion of this evolution pathway._

3. **Identification of linear feature circuits**

    (Convinced)

4. **The authors observe that two groups may differ with respect to sP if module P is active only in one group (and indistinguishable if P is active or inactive in both groups).**

    (Convinced)

5. **Differences arise mainly when a residual predecessor combines with another module, indicating that we might miss other types of causal relations.**

    (Convinced?)

    _I think this may also come from the normalizations and the relatively low cosine similarity between Res - MLP/Attn._

6. **Deactivating a single predecessor causes a greater activation strength drop if it is a group with a single predecessor, which may indicate circuit-like behavior of combined groups.**

    (Convinced)

7. **Different groups react differently to rescaling. Positive rescaling (boosting active features) matters most when residual features mix with MLP or attention. Negative rescaling most strongly affects "From RES."**

    (Convinced)

8. **Multi-layer intervention outperforms single-layer steering of the initial feature set, reducing hyperparameter sensitivity.**

    (Convinced)

9. **Removing topic-related information early allows later layers to recover general linguistic information, aligning with the ability of LLMs to "self-repair" after "damaging" the information processing flow by pruning or intervention into the structure of hidden states.**

    (Relatively Convinced)

    I think the conclusion may be right, _but I think a more controllable experiment is needed, following the contrastive construction method from CAA.

[1] https://arxiv.org/abs/2403.17887
[2] https://openreview.net/forum?id=XAjfjizaKs

**Essential References Not Discussed:**

None.

**Experimental Designs Or Analyses:**

Based on my review, there are some points that could be strengthened or clarified:
1. The paper relies heavily on cosine similarity to map features across layers. The choice of threshold for determining whether two features are related could be further justified. I suggest the authors could explore the sensitivity of their results to different threshold values and provide more analysis on how the threshold was selected.
2. The paper demonstrates successful steering in specific scenarios. However, it would be helpful to explore the generalizability of the steering method to a wider range of tasks and topics.

**Methods And Evaluation Criteria:**

Based on my understanding, the methods and evaluation criteria proposed in the paper seem to make sense for the problem and application at hand. Here's a breakdown:

**Problem:** The paper addresses the challenge of interpreting and controlling the behavior of LLMs. Specifically, it aims to: (1)**Improve interpretability** (2) **Enable steering**.

**Proposed Methods:**
- **Data-free cosine similarity:** This method is used to track the evolution of features across layers. As discussed in Claims And Evidence 1, I think the authors should give more illustrations of the cosine similarity results and analysis of features that are mapped to be the same.
- **Feature flow graphs:** These graphs visually represent how features evolve across layers. This is a helpful tool for providing a clear and intuitive way to understand the complex interactions between features.
- **Multi-layer steering:** This technique leverages the feature flow graphs to control model behavior by intervening on specific features across multiple layers.

**Other Comments Or Suggestions:**

I have no further suggestions to this paper.

**Other Strengths And Weaknesses:**

None.

**Questions For Authors:**

1. Some layers share less cosine similarity with others, e.g., the first layer; does this affect the mapping/steering?
2. More charts/graphs about the dynamics of cosine similarity?
3. Can the co-existence of activation values or the transference be further evidence of the evolution claims?
4. How about the sensitivity of the steering results to different threshold values? And
5. How was the threshold selected?

**Relation To Broader Scientific Literature:**

1. The method provides a straightforward way to identify and interpret the computational graph of the model without relying on additional data.
2. To my acknowledgement, the authors are the first to use SAE features from different layers to control LLM generation.
3. The improved controllability has positive implications for alignment, interpretability, and safe deployment of AI systems, as it can allow developers to steer models away from harmful or biased outputs

**Theoretical Claims:**

I have checked the correctness of proofs about the feature matching, evolution of feature, and activation of theme. I believe they are theoretically correct.

---

> ### Author Rebuttal · Authors · 2025-04-01
>
> Thank you for your insightful comments and questions.
>
> > 1. Some layers share less cosine similarity with others, e.g., the first layer; does this affect the mapping/steering?
>
> We observe that layers indeed vary in feature-matching quality, though we have not yet quantified this systematically (e.g., via similarity analysis of feature descriptions). Your suggestion to assess this further is valuable, and we will incorporate such analysis to strengthen our paper.
>
> Notably, we find that certain layers (e.g., the 4th, 9th, and 18th in Gemma) exhibit differences in how well features match their top-1 similar counterparts in the MLP versus the previous residual stream. This aligns with feature clustering patterns, as shown in Figures 3 and 18b: early layers lack strong cluster separation, leading to poorer matching, while later layers converge toward a single dominant cluster with less distinct groupings.
>
> https://anonymous.4open.science/r/icml_rebuttal-5064/out_mlp_scatters.png
>
> https://anonymous.4open.science/r/icml_rebuttal-5064/out_attn_scatters.png
>
> We have not identified generalizable differences in steering effectiveness across layers, as our analysis so far has focused on a limited set of features and graphs. However, we note that steering MLP or attention features often produces negligible effects, regardless of the target layer or steering coefficient. We hypothesize that this could stem from imperfect matching or the possibility that these features operate differently (e.g., requiring other active features in the residual stream to function). Further investigation is needed here.
>
> > 2. More charts/graphs about the dynamics of cosine similarity?
> >
>
> We have measured the mean cosine similarity between hidden states at layer outputs and three SAE-related positions, plotted against relative layer distance:
>
> https://anonymous.4open.science/r/icml_rebuttal-5064/cossim_models.png
>
> The divergence in Pythia’s results may arise from its parallel residual stream architecture: $\text{Layer}(x)=x+\text{MLP}(\text{norm}_1(x))+\text{Attn}(\text{norm}_2(x))$. We hypothesize that higher cosine similarity between hidden states correlates with better matching quality, though this depends on SAE and model architectures—an aspect we plan to explore further.
>
> These findings may also explain why steering MLP and attention features is less impactful than steering residual features. For additional results on correlation matching and group dynamics in Pythia and GPT-2, please see our response to Reviewer jyW6.
>
> > 3. Can the co-existence of activation values or the transference be further evidence of the evolution claims?
> >
>
> Yes. If you refer to transference across forward layers, we find that matching via the residual stream generally works well, whereas matching with MLP or attention features in the next layer is often poor. We have prepared example graphs of activations for specific texts, available at.
>
> https://anonymous.4open.science/r/icml_rebuttal-5064/g_3_t_0.5.png  (g_3_t_0.8.png/g_3_t_0.85.png)
>
> > 4. How about the sensitivity of the steering results to different threshold values?
>
> The steering effect varies significantly depending on the graph and target feature. For instance, steering the "London" feature barely influences London-related tokens unless the coefficient is very high, at which point it introduces fashion-related terms (see [1] and Appendix E). Some features even show minimal response to steering.
>
> Thus, different graphs react differently, and a systematic study of threshold sensitivity would require extensive experimentation across many features and setups. While we did not aim to optimize thresholds (given their dependence on model/SAE architecture, SAE quality, etc.), we acknowledge this as an important direction for future work.
>
> For our steering experiments:
>
> - Deactivation did not use thresholds, i.e. full graphs from 0th to 25th layers were used.
> - Activation thresholds were chosen empirically: higher thresholds sometimes diluted the theme by including weakly related features, while lower thresholds reduced theme prominence.
>
> > 5. How was the threshold selected?
>
> Initially, we derived thresholds from feature group analysis (Section 5.1) and correlation studies. For example, in Gemma’s 18th layer, we colored features with >0.65 correlation to MLP (orange) or residual (blue) features, then applied linear separation:
>
> https://anonymous.4open.science/r/icml_rebuttal-5064/thresholds.png
>
> For experiments, we determined thresholds $s^{(R)}=0.5$ and $s^{(M, A)}=0.15$ by computing feature activations across texts, building graphs with active features highlighted, and by  manually examining those graphs for threshold values that preserved semantic consistency and co-activation of features in graphs. While expert input remains helpful, we believe this process could be automated in future work.
>
> [1] https://arxiv.org/pdf/2411.02193

---

> > ### Comment · Reviewer_vBgp · 2025-04-05
> >
> > Thanks for your detailed response. Most of my concerns have been addressed. I have revised my assessment accordingly.

---

### Official Review · Reviewer_z5bz · 2025-03-14

**Overall Recommendation:** 3

**Summary:**

The authors introduce a method to allow for cross layer and mulit-module (mlp, residual stream, attention block) level mapping of SAE based features, creating a flow graph using a data-free cosine similarity technique (between feature embeddings and encoder blocks), that allows a person to interpret and trace how specific features persist, transform, or first appear across layers. The paper analyzes such feature evolutions on GemmaScope and LlamaScope SAEs and models and also demonstrates how their method facilitates cumulative layer wise direct steering of model behavior by amplifying or suppressing chosen features for targeted control in text generation and potential causal circuit discoverability.

## update after rebuttal
While the paper provides an interesting method by which to do cross layer, multi-module analysis, and the authors have said they will address the concerns I stated ( which are largely organizational and for clarity),  without seeing the revisions and organization changes ( which will be substantial ) its hard to move the current overall recommendation.

**Claims And Evidence:**

The main methodological proposals/claims made by the paper involve:
1. cosine similarity based feature matching (section 3.2),
2. tracking the evolution of features across modules (mlp, res, att) (section 3.3),
3. 4 heuristics for understanding patterns seen over similarity scores for features/modules (translating, processing, new born and not explained by method) (3.3),
4. discovering long range feature flows by composition of short range matching over consecutive layers (section 3.3)
5. Identification of linear feature circuits /  flow graphs ( section 3.4)
6. Model steering at the flow graph level

The points overall are well motivated and backed up, however points 1, 3, and 5 lack some supporting evidence in the main body of the paper that could help with clarity.

Point 1, section 3.2 is quite clear and easy to follow, however the corresponding results section 5.1 “Identification of feature predecessors” was less so.

(Q1) For Fig 4, what are the raw counts of these groups before looking at differences?  its hard for me to understand 4 in general and that may help, but from this figure i mostly see that most groups similarity scores are stats sig different (with the slight exception of ATT where AB group has lower s(A) , but still above 75%).

(Q2) In Lines 316-19 how can you tell if features are emergent from Fig 5 alone? In terms of “propagating from preceding layers” do you mean because “From Res” goes up?  it might help to have a guide to reading this graph to make it more clear.

(Q3) Lines 326-329 make references to differences in dataset in latter layers, but its hard to tell from graph alone of any differences as they look relatively the same to me?

(Q4) LlamaScope is mentioned in the models section 4.1 and mentioned in one line in section 5.1 (which itself points to Figure 18b in the appendix).   Because its relegated to the appendix, i didn’t notice that the  Gemma and Llama are quite different over all ( see From Res relationship to other lines in both for instance).. How is this difference explained ?   Discussion comparing findings on these two models and their SAEs should be expanded as they seem fairly central to any generalization claims for point 1

(Q5) Throughout, its appreciated you include a “not explained” heuristic ( heuristic D in section 3.3), but discussion on what might be causing “from nowhere” to be the highest scoring pattern observed throughout in Fig 5 and Fig 18b?



Point 3, in 3.3 in describing the heuristics section for phenomena that can be explained, the phrases used are “the feature likely exists …”, “the feature was likely processed ..”, and “the feature may be newborn ….” , and the experiment results assume these heuristic short hands to be true in a way that is difficult to assess.  (Q6) Is there anyway to validate these short hands?

Point 5 ( section 3.4 ) is said to be validated in experiments and Appendix E while the appropriate experiment setup section 4.2 “deactivation of features” which I think allows for casual circuit claims says to look at Appendix A for experimental matching strategies and metrics quantifying effectiveness.  The appropriate results section ( 5.2 deactivation of features ) makes reference to rescaling coefficient r ( not mention in the main body of the paper ).  (Q7) For clarity the appropriate context/setup needs to be introduced in the main body of the work and towards that end the findings related to Figure 7 (which while interesting don’t seem central to the papers main claims) seem like they could be moved to the appendix and you could use the space gained to give more setup for deactivation in 4.2?

(Q8) Its unclear how Figure 10 shows that cumulative intervention outperforms the single layer approach?  It seem reasonable to say that for layers 18 on this true, but not for early ones?  Is my understanding correct?  If not, this should be expanded/clarified.

(Q9) Also in the caption of Figure 10, the orange and blue lines are said to be multi-layer interventions while in the legend of the Fig 10 (RHS) blue is said to be One Layer?

(Q10)  In Figure 11  ( single, constant, exponential, linear ) steering approaches are shown without being explained in the main body of the paper?  Precise details are fine for the appendix, but setup and metrics needed to understand figures in the main body of the paper should be explained in the main section so that the paper is self contained.

(Q11) In reference to lines 420 and 421, “We conclude that multi-layer interventions indeed affect the model more than single-layer approaches “, given my questions on Fig 10 and Fig 11 this is not very clear to me.

Nit:
Line 191, R == R_L-1 was a little confusing as a convention to me. its explained clearly later so maybe just use R_L-1 alone instead?

Overall the method is reasonable and the findings are interesting, but it seems selecting the more important findings to highlight and focus on and moving some of the others out to the appendix for space could benefit clarity.

**Essential References Not Discussed:**

None outside of the 4 month window for concurrent works that I know of.

**Experimental Designs Or Analyses:**

Proposed and evaluation criteria are mostly reasonable.  See Claims And Evidence section on questions and suggestions for how to improve clarity.

**Methods And Evaluation Criteria:**

Proposed and evaluation criteria are mostly reasonable.  See Claims And Evidence section on questions and suggestions for how to improve clarity.

**Other Comments Or Suggestions:**

See Claims And Evidence section for comments/suggestions.

**Other Strengths And Weaknesses:**

This work introduces a novel interpretable data-free method for multi-layer steering, which enables the tracking of concept evolution across layers and the identification of computational circuits through targeting the weights of pretrained SAEs.  While there are some issues with clarity, overall the method could allow for expanded analysis of models leveraging SAEs at th moment.

See Claims And Evidence section on some improvements needed for clarity and setup of some experiments.  Future work would need to address newer work (https://arxiv.org/abs/2501.17727)  showing SAEs can interpret randomized vectors and the need to tie them to downstream performance in order for them to be grounded.

**Questions For Authors:**

See Claims And Evidence section on questions

**Relation To Broader Scientific Literature:**

This work introduces a novel interpretable data-free method for multi-layer steering, which enables the tracking of concept evolution across layers and the identification of computational circuits through targeting the weights of pretrained SAEs.

**Theoretical Claims:**

N/A

---

> ### Author Rebuttal · Authors · 2025-04-01
>
> Thank you for your careful reading and insightful questions.
>
> ---
>
> **Q1.** The purpose of Figure 4 is indeed to show that these groups differ significantly. Below are the raw counts of elements in each group:
>
> | Nowhere | RES | MLP | ATT | RES & MLP  | RES & ATT | MLP & ATT | RES&MLP&ATT |
> | --- | --- | --- | --- | --- | --- | --- | --- |
> | 3396295 | 2338643 | 972662 | 574635 | 769450 | 606080 | 261068 | 308578 |
>
> See Figure 13 for an evaluation of how frequently groups intersect. Those values are calculated as $|G_1\cap G_2|/|G_1|$, where $G_1$ is a row group and $G_2$ is a column group.
>
> **Q2.** Figure 5 alone cannot confirm feature emergence. We instead analyze group proportions across layers. Features are classified as *emergent* if closer to module (MLP/ATT) features than residual stream features (Figures 12b, 18c), placing "From MLP", "From ATT", and "From MLP & ATT" in this category. *Translated* features are those closer to the previous residual stream ("From RES"). See also our answer to Q6.
>
> **Q3.** Comparing FineWeb and Python Code in the later layers reveals a higher presence of “From nowhere” features than “From RES,” distinguishing this dataset from others. The same trend appears for Llama (see Figure 18b) and Pythia-70M (mentioned in our response to reviewer jyW6).
>
> **Q4.** We may have misparsed your comment—did you mean "I did notice"? Model differences exist but stem from varying architectures, parameter counts, and SAE training procedures. A full explanation requires deeper study (e.g., identically architected SAEs on shared data). We will provide additional discussion and experiments on Pythia and GPT-2 in the revised manuscript to address this further.
>
> **Q5.** We believe there are two primary causes for features labeled “From nowhere.” First, there may be a matching error—artifacts from SAE training or a situation where the true predecessor is not the top-1 but perhaps the top-2 or top-3 similar feature. Second, some features could be combinations of two or more features, making them harder to detect via a simple top-1 match; they could also transform (e.g., rotate) as they pass through the layer, as reviewer jyW6 pointed out. Using top-5 cosine similarity, as in our deactivation experiment, reduces the “From nowhere” group substantially, but it is unclear if it can ever be made negligible. Expanding the SAE dictionary might also risk increasing false positives under the top-1 strategy, so this trade-off requires more investigation. We will extend our discussion of the “From nowhere” group to clarify these points in the revised version.
>
> **Q6.** We currently think of two general approaches for validating these shorthands.
>
> The first involves analyzing their semantics and activation patterns. For instance, if a target feature activates on tokens like *“the, a”*, while its predecessor in the residual stream fires on *“the, The”* and the corresponding MLP feature activates on *“a, A, an”*, this suggests the target feature emerges from their interaction—we categorize it as *“processed.”* Features with nearly identical descriptions and activation patterns across layers likely belong to the *“translated”* group, indicated by high $s^{(R)}$ values. Conversely, features without semantic ties to the residual stream but aligned with MLP or attention counterparts may be *“newborn”*—new linear directions introduced by those modules. Expanding this analysis beyond top-1 matches (e.g., to top-5) could further enhance its reliability.
>
> Figures 3 and 18b illustrate clusters in the top-left corner, where features lack $s^{(R)}$-similar predecessors in the residual stream. This could imply either new feature introduction by the MLP (e.g., stored knowledge) or transformation of an initially dissimilar residual feature via MLP processing. In contrast, Figure 5 shows a clear predominance of the ”From RES*”* group, suggesting most features propagate across layers with minimal modification.
>
> While this analysis could be automated, it hinges on either reliable feature explanations or raw activation data.
>
> The second approach tests feature roles through steering experiments. For example:
>
> 1. Steering only the target feature,
> 2. Steering its residual predecessor, or
> 3. Steering the residual predecessor while deactivating active MLP/attention predecessors.
>
> Comparing these setups could reveal mechanistic differences, though we have not yet implemented this systematically. Preliminary versions of (a) and (b) appear in Section 5.3, though (b) in our current work involves steering the entire flow graph, not just the residual stream.

---

> > ### Comment · Reviewer_z5bz · 2025-04-03
> >
> > Thank you for clarifying and answering most of the questions I've listed ( Q1- Q6).  Point 5 (Q7) is particularly important for the overall clarity of the paper so I hope you'll be able to address that.

---

> > > ### Author Response · Authors · 2025-04-04
> > >
> > > We sincerely apologize for the inconvenience regarding the remaining questions: we encountered issues with formatting the rebuttal in time, resulting in it being cut short. Below we briefly answer other questions and outline our plan considering the improvements of clarity and presentation:
> > >
> > > **Q7.** We agree that the main body should contain those details and will improve the structure by making the following improvements:
> > >
> > > - We will extend 4.2 by a) introducing the rescaling coefficient, b) briefly describing matching strategies and c) explaining our activation change metric; considering steering experiment, we will d) clarify the single-layer/cumulative terminology, e) the baseline approach we compare our method with, f) explain our evaluation strategy and g) briefly describe the three cumulative steering approaches (constant, linear, exponential).
> > > - As you rightfully pointed out, results and text regarding Figure 7 would be better placed outside the main text, and we will move them into Appendix C.2.
> > > - We will improve conciseness of the Results section by removing redundant explanations that should be given in 4.2, such as the first paragraphs of 5.2 and 5.3.
> > >
> > > **Q8 & Q11.** Cumulative steering outperforms single-layer steering in terms of requiring lower rescaling coefficients, which improves stability, while single-layer steering, as indicated by the total score, can be more effective overall but demands either spanning multiple layers with a reliable graph or selecting particularly impactful layers, which introduces additional hyperparameter dependency. Thank you for highlighting this, we will revise these descriptions for clarity, both in the Results and Discussion section.
> > >
> > > **Q9.** By “multi-layer,” we intended to encompass both cumulative and single-layer approaches (since we can potentially find related features in other layers), as opposed to *single-feature* steering. We would also clarify that.
> > >
> > > ---
> > >
> > > We appreciate the time and effort you dedicated into reviewing our paper. We will gladly incorporate results of this discussion and improve the clarity and presentation of our paper, and we hope that we have addressed your concerns and you find our responses satisfactorily. If so, we would be deeply grateful for your support of our paper. Thank you again for your thoughtful review.

---

### Official Review · Reviewer_jyW6 · 2025-03-17

**Overall Recommendation:** 2

**Summary:**

The paper investigates whether features in LLM activations identified by SAEs trained on individual activation locations - namely residual streams, attention outputs and MLP outputs - can be linked to one another, so that we can, broadly speaking, try to answer questions like "where did a feature come from?" and "how did a feature evolve through the layers?".

Features across SAEs are linked to each other via the cosine similarity of their decoder vectors. Specifically, given:
- a residual stream SAE feature $f$ in layer $L$;
- SAEs trained on the previous residual stream of layer $L-1$, as well as the MLP and attention outputs in-between;

we define $s^{(R)}, s^{(M)}, s^{(A)}$ to be the top cosine similarities of $f$ with any decoder vector in the previous resid (R), MLP (M) and attention (A) SAE decoder vectors. The relative magnitude of these values is used to classify a feature as being one of:
- translated from the previous residual stream without MLP/attn involvement;
- processed by the MLP or attn when all values are high
- "newborn", created by the MLP or attn
- unexplained when all values are low.

The main body experiments aim to answer the following questions:
- **feature predecessors**: this checks that feature predecessors identified using the paper's method correlate with SAE feature activation, i.e. if the predecessor of $f$ activates on a datapoint, does $f$ activate too?
- **deactivation of features**: if we deactivate a predecessor feature of $f$ by subtracting it from the activation, will this lead to $f$ not activating?
- **model steering**: can we improve upon naive single-layer steering by steering multiple features across layers jointly when they're identified as connected via predecessor relations?

The results suggest the following:
- we generally observe a clear cluster of features that have a strong predecessor in the preceding MLP layer but not in the previous residual stream layer, and for these features it's also generally the case that if the feature is active, its MLP predecessor is also active. By contrast, for attention we see no such pattern.
- feature deactivation guided by cosine similarity performs significantly better than a baseline using random choice. Deactivation seems to have higher impact on activations when done using residual stream predecessors vs MLP/attn predecessors.
- for steering to activate a certain topic in the generation, results don't clearly show that the proposed methods outperform baselines or single-layer steering. For steering to de-activate a topic, i.e. suppress mentions of it, there seems to be some benefit to multi-layer steering versus single-layer.

## Update after rebuttal
The rebuttal has not meaningfully changed my assessment of the paper. Despite other issues, I hesitate to recommend acceptance mostly because I don't see a strong motivation behind the problem being studied and don't see the usefulness of the results to the broader field of interpretability.

**Claims And Evidence:**

In general, the claims made in the paper are supported with evidence reasonably well, however there are some issues with the methodology and presentation of results (described below and in subsequent sections) that make it difficult to evaluate the scope and novelty of the findings.

- a key property of the methods in this paper is that they can only tell us about how the same feature persists (or not) through layers, or when it "appears".
	- This limits the conclusions that can be drawn from the results; by contrast, a much more interesting (and more difficult) question would be to understand how different features combine through attention and MLP blocks to create *new* features.
	- Furthermore, the approach in this paper fails to fully account for the possibility that features "rotate" across layers, as described e.g. in https://transformer-circuits.pub/2024/crosscoders/index.html; to some extent, this is mitigated by linking features through successive layers, but it is unclear if this overcomes the problem. I think that a careful study using a tool similar to cross-coders, as well as comparing feature activations over datasets of texts, could help answer this question.
- in general, the methods and claims may be shedding more light on the geometric structure of the SAE matrices as opposed to the workings of the model. For instance, it has been observed that SAE encoder and decoder vectors for the same feature have substantial though not perfect cosine similarity (see e.g. https://transformer-circuits.pub/2023/monosemantic-features#comment-nanda). This alone *may be* enough to explain the results on predecessors and deactivation!
- in general, the paper would have benefited from more methods and experiments that establish the relevance of the phenomena explored to the end-to-end behavior of the model, as opposed to internal geometric structure that may be an artifact of SAE optimization. The steering experiments are one promising step in this direction.
- it should be noted that while the cosine similarity method is advertised as "data-free", the features we're taking the cosine similarity of themselves came from SAEs, which are trained on vast amounts of data. So strictly speaking this is not a data-free interpretability method (as opposed to a method that is purely based on the weights of the model). In general, "data-free" interpretability methods have the promise of explaining the out-of-distribution behavior of a model precisely because they are independent of any properties of the data apart from what is encoded in the model weights, which is a key motivation for considering such methods. However, this motivation does not really apply to the methods in this paper. However, it should be noted that truly data-free methods have so far struggled to provide useful interpretability insights.

**Essential References Not Discussed:**

- in the discussion of the topk activation function, a foundational reference that for the first time proposed the use of this activation in SAEs and established its improved metrics and scaling properties is missing: Gao, L., la Tour, T.D., Tillman, H., Goh, G., Troll, R., Radford, A., Sutskever, I., Leike, J. and Wu, J., 2024. Scaling and evaluating sparse autoencoders. _arXiv preprint arXiv:2406.04093_.

**Experimental Designs Or Analyses:**

- In general, for many of the experiments and figures there is insufficient detail describing what is being done/visualized.
- I don't quite follow Figure 7. Is "activation change" measuring the drop in the SAE feature's pre-activation? Also, if we're deactivating 1 predecessor at a time, where is it shown how the effect evolves with the number of predecessors deactivated? Also, how do we get multiple predecessors for a feature? Do we just look at the top k highest cosine similarities?

**Methods And Evaluation Criteria:**

Some shortcomings:
- there is a lack of a baseline approach for linking features to one another across layers that the approach of the paper can be compared to. The top candidate that comes to mind is an activation-based metric that correlates feature activations over a large set of examples.
- in general, there is often not enough detail to get a full picture of how a given method is implemented.
- a potential source of noise for the given methodology is that it is known that SAEs trained on the same activations can arrive at different features with different random seeds. See the paper "Sparse Autoencoders Trained on the Same Data Learn Different Features" by Paulo and Belrose. In the absence of any "anchor" between different layer SAEs, linking features in such a naive way may be suboptimal; a strong activation-correlation baseline is desirable.
- the way in which behavioral and coherence scores are assigned in the steering experiments is not explained.
- combining behavioral and coherence scores in a single metric is a potential source of misconceptions and illusory results. Furthermore, doing so via multiplication is not very principled: what's the interpretation of a unit in the resulting scale? It feels overly confusing to think about that. A very high behavioral score can be achieved at moderate coherence if the model repeats a phrase related to the topic. In general it's more insightful to look at both behavioral & coherence metrics together on a 2D plot.

**Other Comments Or Suggestions:**

- Section 3.1. feels somewhat redundant, given the introduction which already establishes the motivation? Consider adding a brief sentence to the introduction citing the main relevant works describing the motivation, and cutting this section.
- I think the exposition would flow better if it is stated upfront that we assume that the decoder vectors have unit norm (lines 139 and 140, right column)
	- Sections 3.4 and 3.5 are quite vague in terms of methodology; are they really necessary? I don't feel like I got much out of them beyond "we do some cool stuff". Maybe cut & incorporate in later sections that flesh out these ideas?
	- Same applies to some extent for 4.2.; I wonder if presentation would be better if there was a single place in the paper that has a 1-sentence brief intuition for each experiment + a very concrete description?

**Other Strengths And Weaknesses:**

Weaknesses:
- The presentation of the methods is confusing (also see comments/suggestions section below). In particular, the "Results" section is actually the one that describes the methods in sufficient detail, whereas previous methods sections only vaguely point at what's going to be done.
- You say "We found that cosine similarity between decoder weights is a valuable similarity metric, and we focus on this approach." (line 160), but this is never justified in terms of alternative methods and metrics. It would be a valuable addition to the paper.
- You say in 3.4. that your method can detect when MLP or attn *remove* features from the residual stream, but there's no evidence for what removal would look like in the paper?
- The 4 subplots of Figure 5 are basically identical - maybe just include 1 in the main body, and say that for other datasets it looks basically the same. Furthermore, this suggests that these lines are more a geometric property of the ensemble of SAEs as opposed to a data-specific property.

**Questions For Authors:**

- it is customary to subtract the decoder bias $b_{dec}$ from $h$ before applying $W_{enc}$ (cf line 77, left column). Omitting this changes the architecture of the SAE, though in principle the bias can be absorbed in the encoder bias by changing $b_{enc}$ to $b_{enc} -W_{enc}b_{dec}$.
- what is the matrix norm in line 87 (right column)? I assume Frobenius
- in 2.3., is the intention that $B<A$?

**Relation To Broader Scientific Literature:**

It's hard to situate the contributions of the paper in the broader literature because of the lack of baselines and the possibility that many of the results follow from known facts about SAE encoder/decoder geometry. The exception is the steering experiments, which may be a useful addition to the literature, especially if presented in more detail / with easier to interpret analyses (see previous points).

**Theoretical Claims:**

N/A

---

> ### Author Rebuttal · Authors · 2025-04-01
>
> Thank you for your valuable and extensive review. We aim to address your concerns below.
>
> **Claims And Evidence**
>
> Our work examines SAEs at various points in the model and linking their features allows us explore its computational structure—such as the technique described in Appendix F, which mimics a transcoder approach. Thus, we can shed more light on the model’s behavior than just geometry. Permutations [2], top-1 correlation [3], and top-1 cosine similarity (our work) are all special cases of this feature mapping approach.
>
> We respectfully disagree with labeling our approach as non-"data-free", since we use this term following the prior work [2]. While SAEs could be trained on the LLM’s data, we solely analyze model weights, not external data, justifying the "data-free" description.
>
> **Methods And Evaluation Criteria**
>
> We appreciate your suggestion for a data-driven baseline. While useful, such methods struggle with sparse SAEs, demonstrating the advantage of our data-free approach where adjusting k in top-k matching can better address these limitations.
>
> We tested Pearson correlations on 100K non-special tokens from the FineWeb “default” subset for each feature in Gemma Scope’s even layers and all layers of Pythia-70M-Deduped and GPT-2. Using 500 samples (instead of 250) as described in Appendix A.1, we identified feature groups.
>
> https://anonymous.4open.science/r/icml_rebuttal-5064/gemma_corr_cos.png
>
> https://anonymous.4open.science/r/icml_rebuttal-5064/pythia_corr_cos.png
>
> https://anonymous.4open.science/r/icml_rebuttal-5064/gpt_corr_cos.png
>
> Correlation-based matching reduced the "From nowhere" group and better identified attention module predecessors, although the mismatch with Gemma Scope SAEs still lowered quality. For Pythia, results aligned closely with Llama Scope, reflecting clearer attention features.
>
> However, correlation-based matching did not consistently outperform top-1 cosine similarity and performed worse on out-of-distribution Python code (further from FineWeb). Top-1 cosine and top-k correlation predecessors showed strong agreement for Gemma Scope and GPT-2 residual SAEs but weaker alignment for module-based SAEs, consistent with prior feature propagation findings.
>
> These results broadly characterize correlation-based performance. We plan to include these comparisons but welcome requests for specific details to enhance our discussion.
>
> Behavioral and Coherence scores follow the setup from [1], with details and the system prompt described in Appendix B. In brief, we ask a model to assess whether a specific theme is present (Behavioral) and to rate the text’s language quality (Coherence), assigning each an integer score from 0 to 5. These scores are then normalized to the [0, 1] range. We acknowledge that this explanation belongs in Section 4.2 and will revise it accordingly.
>
> As you noted, multiplying the scores ensures that moderate Coherence and high Behavioral (or vice versa) results in a moderate overall score as expected. Both scores are illustrated in Figure 9.
>
> **Experimental Designs Or Analyses**
>
> We define the activation change as $1-\mathbf{z}^{new}/\mathbf{z}^{old}$, where $\mathbf{z}$ is a feature activation after applying JumpReLU.
>
> Each feature may have up to three predecessors (residual, MLP, attention). To identify active predecessors, we use four methods (Appendix A.2).
>
> For features with active predecessors in both residual (R) and MLP (M), we label them “From RES & MLP” and run forward passes deactivating R, M, or both (“deactivated one at a time”). In top-k method, up to five features may be deactivated per predecessor (though few are typically active).
>
> Figure 7 subplots group results by predecessor type; bars show deactivation effects (e.g., deactivating “mlp” in “From RES & MLP” yields ~0.25 mean activation change).
>
> **Other Strengths And Weaknesses**
>
>
> While we did not explicitly justify cosine similarity as a metric early on, Section 5.2 and Appendix F compare it with a permutation-based method.
>
> By “removal” or “suppression,” we refer to specific linear combinations of features (e.g., “king – man”), similar to combinations that create new semantics (e.g., “woman + power”). We suspect these combinations are common.
>
> **Questions For Authors**
>
> We use the L2 norm, treating vectors as sliced decoder columns.
>
> While our presentation assumes B < A,  this is not strictly necessary.
>
> Thank you again for your valuable feedback. We will reorganize our paper, improve clarity, and add the missing reference. We hope these clarifications sufficiently resolve your concerns and encourage you to reconsider your evaluation. Thank you for your time, and please let us know if you have any further questions or requests.
>
> References:
>
> [1] https://arxiv.org/pdf/2411.02193
>
> [2] https://arxiv.org/pdf/2410.07656
>
> [3] https://arxiv.org/abs/2410.08869

---

### Decision · Program_Chairs · 2025-05-01

**Decision:**

Accept (poster)

**Comment:**

The authors introduces a method for analyzing and manipulating the evolution of SAE-derived features across layers in large language models. By aligning sparse autoencoder features using cosine similarity and constructing flow graphs, the authors provide a framework for tracking feature propagation, identifying circuit-like structures, and performing multi-layer model steering. The experiments show that interventions guided by these flow graphs can enhance or suppress specific themes in generated text while preserving coherence.

The motivation is well grounded, and the use of data-free methods for inter-layer feature matching is a strength. The paper also makes a solid empirical case that steering based on multi-layer feature tracking can improve control over model generations, with applications in interpretability and alignment. That said, some reviewers raised concerns about novelty and generality. While the results are promising, the feature matching and steering procedures extend known ideas (e.g., cosine similarity and SAE-based control) rather than introducing fundamentally new techniques. Evaluation is primarily focused on a single model family, and it’s unclear how broadly the results generalize.

After considering the reviews and paper, I believe the positives outweigh the negatives for this paper and believe the work advances our understanding of cross-layer interpretability and model control.  I recommend the authors take the reviewer comments and my notes above into account in their revisions.